

# The coherence of the oceanic heat transport through the Nordic seas: oceanic heat budget and interannual variability

Anna V. Vesman [1,2], Igor L. Bashmachnikov[1,3], Pavel A. Golubkin[3], Roshin P. Raj[4]

[1]Saint-Petersburg State University, Department of Oceanography, Saint-Petersburg, 199034, Russian Federation
[2]Arctic and Antarctic research institute, Atmosphere-sea ice-ocean interaction department, Saint-Petersburg, 199397, Russian Federation;
[3]Nansen International Environmental and Remote Sensing Centre, Saint-Petersburg, 199034, Russian Federation;
[4]Nansen Environmental and Remote Sensing Centre, Bjerknes Center for climate Research, Bergen, N-5006, Norway

*Correspondence to:* Anna V.Vesman (anna.vesman@gmail.com)

**Abstract.** Atlantic Water is the main source of the heat and salt in the Arctic. On the way to the Arctic Ocean via the Nordic Seas, it interacts and mixes with other water masses which affects sea ice extent and deep water formation. The Atlantic Water heat transported into the Nordic Seas has a significant impact on the local climate and is investigated here along with its inter-annual variability using the ARMOR3D dataset, which is a collection of 3D monthly temperature, salinity and geostrophic velocities fields, derived from in situ and satellite data on a regular grid since 1993. The study region includes the eastern part of the Nordic seas, i.e., seven latitudinal transects from Svinoy section (65° N) to the northern part of the Fram Strait (78.8° N). The Atlantic Water heat advection decreases northwards, as a significant amount of heat is lost to the atmosphere and due to mixing with surrounding waters. As observed, the imbalance of heat fluxes in the upper layer leads to an increase in the upper ocean mean temperature over most of the study region. The correlations of the interannual variations of the advective heat fluxes rapidly drop from Svinoy to Jan Mayen sections and between Bear Island and Sorkapp sections. This is a result of a differential damping of periodicities (the 2–3 year and 5–6 year oscillations), as well as of different signs of the tendencies over the latest decades. The heat fluxes at all sections show a consistent change with meridional (C) and western (W) weather types, which is due to the different direction of the Ekman pumping associated with each of the weather types. A certain link to the NAO, AO and EA atmospheric indices is observed only at the southern sections.

**Key words:** the Nordic Seas, the Atlantic Water, heat flux, long-term variability, wavelet analysis

## 1. Introduction

The Arctic region is undergoing significant changes in XX – XXI centuries. Among the other factors, the interannual variations of temperature, sea ice extent and etc. in the Arctic region are linked to the variability in the poleward transport of heat by the





ocean and the atmosphere, often coupled (Jungclaus and Koenigk, 2010; Schlichtholz, 2011, Bucklay and Marshall, 2016;
Bashmachnikov et al., 2018). A significant amount of the northward directed oceanic heat is released in the Nordic and the

Barents seas and in the Whaler's Bay north of Spitsbergen (Piechura and Walczowski, 2009; Moore et al., 2012; Smerdsrud
et al., 2013; Bosse et al., 2018). Previously, the subsurface Atlantic water (AW) was believed to not affect the Arctic climate
after submerging and entering the Arctic (Lenn et al., 2009; Sirevaag and Fer, 2012; Rudels et al., 2013). However, recent
studies show that the AW can reach the upper mixed layer in the Atlantic sector of the Arctic which can be associated with the
recent warming of the Arctic, thus becoming an important factor for the Arctic climate change (Schlichtholz, 2013; Tverberg

et al., 2014; Carmack et al., 2015; Polyakov et al., 2017).

Warm and saline AW is transported north across the Nordic Seas to the Arctic along the continental margin of Norway by the
Norwegian Atlantic Slope Current (NwASC) and along the Jan Mayen Fracture zone and Mohn–Knipovich ridges by the
Norwegian Atlantic Front Current (NwAFC) (Poulain et al., 1996; Orvik and Niiler, 2002; Skagseth et al., 2004). There is
practically no AW transport from the Barents Sea to the Arctic Ocean (Smerdsrud et al., 2013; Mahotin and Ivanov, 2016);

most of the oceanic heat enters the Arctic Ocean through the Fram Strait (Rudels, 1987, 2015;Schauer et al. 2004; Beszczynska-
Möller et al., 2012). The West Spitsbergen Current (WSC), a continuation of the NwASC, has a complex structure near the
Fram Strait, where it is split into several branches and recirculations (Aagaarda et al., 1987; Gascard et al., 2011; von Appen
et al., 2015). Two main paths, the Svalbard branch along the Spitsbergen slope (limited by the 400 meters isobaths) and the
Yermak branch along the western flank of the Yermak Plateau, enter the Arctic, while the recirculation pattern turns

southwestwards, back to the Nordic Seas (Saloranta and Haugan, 2004; Cokelet et al., 2008; von Appen et al., 2015).

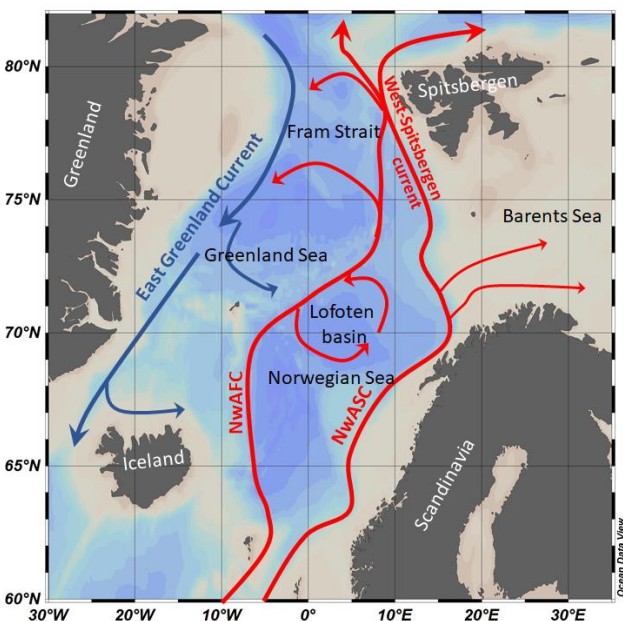

Figure 1. Schematic map of oceanic circulation in the study region, NwASC  - Norwegian Atlantic Slope Current (NwASC),
NwAFC  - Norwegian Atlantic Front Current.





On its way through the Nordic Seas to the Arctic Basin, the AW undergoes dispersion in several recirculations and density

transformation through heat loss to the atmosphere and mixing with surrounding waters (Chafik et al., 2016; Polyakov et al., 2017; Muilwijk et al., 2018; Bosse et al., 2018). The Faroe, East Icelandic and West Icelandic currents carry a total of about 8.0–9.0 Sv, which merge into the Norwegian Current (the NwAFC, the NwASC and the Norwegian Atlantic Coastal Current - NwACC) (Fig.1 1) (Dickson et al., 2008; Rossby et al., 2017). A total of 260–300 TW (reference T = 0 ° C) of heat is brought from the mid-latitude Atlantic to the Norwegian Sea located in the eastern Nordic Seas (Hansen et al., 2008; Rossby et al.,

2017). Across the Svinoy Section located further north in the Norwegian Sea, the Norwegian Current carries an average of 4.0–6.0 Sv (Mork and Skagseth, 2010) and around 150 TW of heat (the estimates vary from 100 to 200 TW, Skagseth et al., 2008 ; Bacon et al., 2015). Therefore, approximately half of the incoming AW heat is released to the atmosphere or heats the Arctic waters coming from the Greenland Sea, even before reaching the Lofoten Basin of the Norwegian Sea known as a region with large winter heat loss (Segtnan et al., 2011).

The NwACC and a part of the NwASC enter the Barents Sea along the northern shelf of Scandinavia as the Nordkapp and Murmansk (Norwegian Coastal) currents with a total average transport of 2.0 Sv (from 1.0 to 3.0 Sv; Smerdsrud et al., 2013); the average annual flow of oceanic heat into the Barents Sea (reference T = 0 °C) is around 50 TW (from 30 to 60–70 TW, Skagseth et al., 2008; Smerdsrud et al., 2010; Skagseth et al, 2011; Bashmachnikov et al., 2018). From 1998, a monotonous increase in the average heat flux (1.5 TW per year) is observed, which is associated with an increase in the volume transport,

rather than temperature of the AW (Schauer et al., 2008; Kalavichchi, Bashmachnikov, 2019).

The total flow through the Fram Strait to the north of the West Spitsbergen Current (WSC) is 6.0–11.0 Sv with a characteristic inter-annual variability of about 5.0 Sv (Schauer et al., 2004, 2008; Fahrbach, 2006; Beszczynska-Möller et al., 2012; Rudels et al., 2013). The difference in the estimates is due to a complex flow structure and difficulty in evaluating the strong recirculations in the Fram Strait. Using a reference temperature of 1.0 °C, the mean heat flux to the north of Spitsbergen during

latest decades was estimated to be 30–40 TW (Schauer et al., 2008; Fahrbach, 2006, Schauer and Beszczynska-Möller, 2009; Rudels et al., 2013) and was found to increase since 1980 (Dickson et al., 2008). Along the WSC the overall transport increases from 3.0 to 8.0 Sv at the southern Fram Strait. Considering the AW inflow to the Arctic of about 2.0 Sv (Beszczynska-Möller et al., 2012), the intensification of the WSC is fed by the entrainment of the Greenland Sea Water (Walkzowski, 2014). The total northwards heat flux through the Fram Strait is around 30 TW (Walkzowski, 2014), i.e., only 10% of the total heat

entering the Norwegian Sea reaches the Fram Strait.

During the recent decades, the time-series of the AW temperature in the Fram Strait and the Barents Sea Opening show a prominent long-term positive trend in the AW core temperature (around 1° C per decade in the WSC) (Schauer et al., 2008), as well as interannual fluctuations with the characteristic periods of 5–6, 8–10 years (Skagseth et al., 2008; Vesman et al., 2017; Muilwijk et al., 2018; Bashmachnikov et al., 2018). The volume and heat fluxes are re-distributed between the Barents

Sea and the Fram Strait, governed by the regional wind patterns through variations of the sea-level anomalies (Lien et al., 2013).

In this paper we analyze the space-time variations in advective heat fluxes along the pathways of the AW into the Arctic.



## 2. Data and Methods

### 2.1 ARMOR3D dataset

The latest version of the ARMOR3D dataset used in this study is a collection of global gridded monthly 3D fields of temperature, salinity and geostrophic currents based on in situ and satellite observations at standard depth-levels and with 0.25° x 0.25°spatial resolution. The data from 1993 are available through the CMEMS web portal (Verbrugge et al., 2017). Joint analysis of satellite (sea-level and sea surface temperature (SST) anomalies) and sub-satellite historical data through a multiple linear regression provides temperature and salinity values on a regular grid at different depth levels. These "synthetic"

temperature and salinity profiles are combined with historical data in the optimal interpolation procedure to obtain the final monthly 3D thermohaline fields (Guinehut et al., 2004, 2012). Geostrophic currents are calculated by extrapolating the sea-surface altimetry currents downwards using the thermohaline fields of the previous step and the thermal wind equations (Mulet et al., 2012).

### 2.2 Mooring data


To validate the volume and heat fluxes derived from ARMOR3D, data from the moorings deployed in the Fram Strait by Alfred Wegener Institute (Beszczynska-Möller et al., 2012, 2015) were used. The dataset consists of the temperature, salinity and currents speed information from 10 moorings stations deployed along 78.8° N from 8.70° E to 2.10° W during 1997-2011. The datasets are available online from PANGEA database(Beszczynska-Möller et al., 2015). All available precalculated

oceanic heat fluxes data from NACLIM project for the Hornbanki station located at 66.50° N 21.30° W (Jonsson and Valdimarsson, 2012) were also used.

### 2.3 Atmospheric data

The ocean-atmosphere heat exchange, as well as short/long–wave radiation balance is derived from the ERA–Interim

reanalysis (Dee et al., 2011) distributed by the European Centre for Medium-range Weather Forecasts (ECMWF). Turbulent heat fluxes across the upper boundary, i.e., to/from the atmosphere, were calculated using the COARE 3.5 algorithm. COARE 3.5 is a modified version of the COARE 3.0 algorithm (Fairall et al, 2003) based on the CLIMODE, MBL, and CBLAST experiments (Edson et al., 2012).



### 2.4 Atmosphere circulation indexes

The North Atlantic Oscillation (NAO), Arctic Oscillation (AO) and East Atlantic (EA) indices were obtained from the NOAA National Weather Service Climate Prediction Center. The statistical links between the oceanic heat transport and the typical atmospheric pressure patterns, characterized by these indices, were estimated.

In 1933, Vangengeim suggested a set of indices characterizing atmospheric circulation. He introduced the concept of an elementary synoptic process (ESP). ESP was understood as the process during which, within the Atlantic-European sector, the geographic distribution of the sign of anomalies of the pressure field and the direction of the main air transportations are preserved. ESP could be generalized in three main types of atmospheric circulation: - the western (W), - the eastern (E) and meridional (C) (Girs, 1978; Prokhorova and Svyashchennikov, 2016). During type W, zonal components of the air circulation at mid-latitudes are strengthened and meridional are weakened. This type of circulation leads to a significant reduction in the interactions of the air masses between the tropics and high latitudes. During the circulation pattern of type C, the Icelandic and Aleutian Lows are practically nonexistent due to development of the high-pressure anomaly over the north Atlantic, the so called Atlantic Ridge. Further east, the Siberian Anticyclone strengthens and becomes connected with the Polar Anticyclone. Type E features strong meandering of the mid-latitude jet, as in C, but the main high-pressure ridges change to troughs and vice versa. In this type, the Icelandic Low is well developed and the stationary anticyclones are observed over Europe and America (Bezuglova & Zinchenko, 2009). Vangengeim – Giers classification helps to highlight variations of the wind patterns over the study region, only partly captured by the atmospheric indices above.

The Atlantic Multidecadal Oscillation (AMO) index, shaping the long-term variability of water temperature in the tropical to mid-latitude North Atlantic, i.e. the temperature of the waters entering the Nordic Seas, was also downloaded from NOAA National Weather Service Climate Prediction Center.

### 2.5 The study region

The transects for calculation of the oceanic advective heat fluxes are drawn across the main pathways of the AW in the Nordic Seas, from the latitude of the Svinoy section at 65° N to the northern part of the Fram Strait at 78.8° N (Fig. 2). The position of the transects may have a significant effect on the absolute values of the heat flux estimates. The sections were drawn to be approximately perpendicular to the direction of the mean currents, i.e. to the continental slope and the underwater ridges, as the currents are strongly bottom trapped. The continental shelf was assumed as the eastern boundary for most of the sections. The selection of the western limits of the zonal section is an ambiguous task. In this study the western limits correspond to a point with the minimum velocity of the NwAFC, before the sign of the mean meridional flow is reversed (Fig. 2). This minimizes the effect of the return flow and an unstable part of the jet flow dominated by eddies (Bashmachnikov et al., 2020).



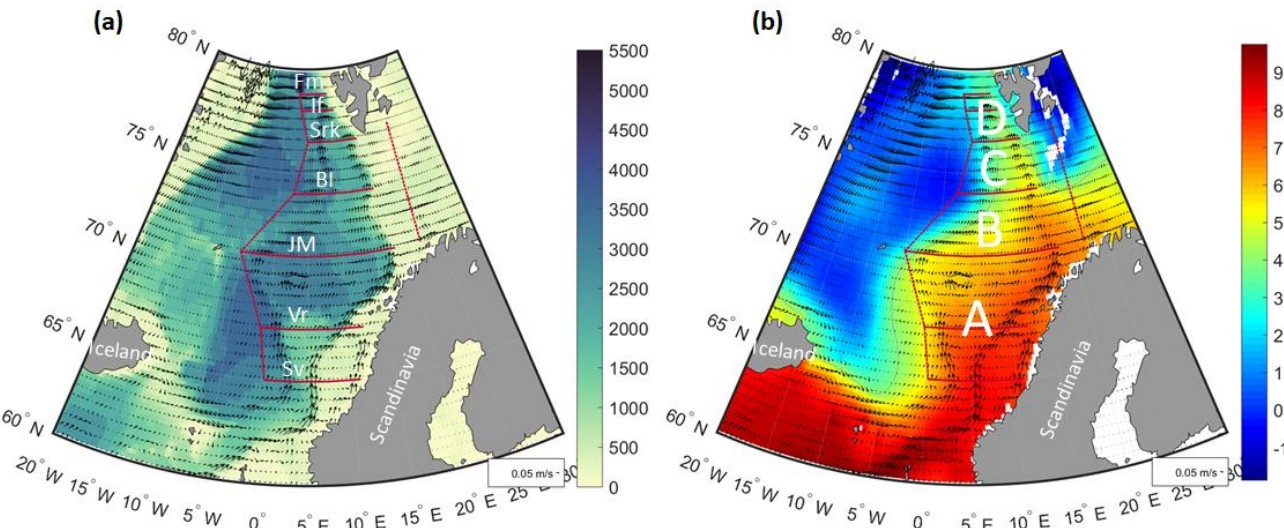

Figure 2. Transects used for the calculation of the oceanic heat fluxes and areas for calculations of the vertical heat fluxes (A-D). (a) represents the bathymetry (m), (b) - mean water temperature on 50 m ($^{o}$C). Black arrows indicate the mean currents. The sections: Sv - Svinoy, Vr - Voring, JM - Jan Mayen, BI - Bear Island, Srk – Sorkapp, If – Isfjord, Fm – Fram; the areas: A – from Svinoy to Jan Mayen, B – From Jan Mayen to Bear Island, C - from Bear Island to Sorkapp, D – from Sorkapp to Fram

Despite minor variations in the absolute values of the heat fluxes, when varying the eastern and western limits, the seasonal and interannual variability of the fluxes remains unchanged (see example in Fig. 3).

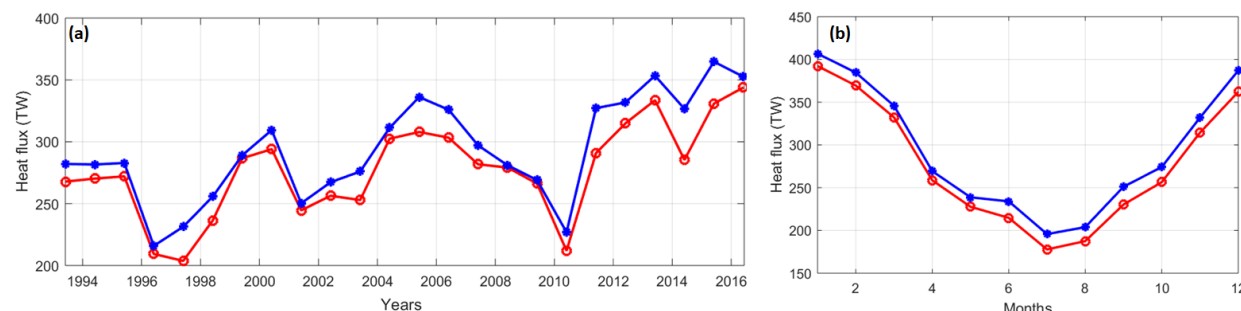

Figure 3. a - interannual and b - seasonal variability of the heat fluxes across Jan Mayen section calculated using different western and eastern boundaries (from Norwegian shore to the Jan Mayen island (blue) or limited by the shelf on the east and the current minimum on the west (red), see Figure 2)

The heat fluxes through the western boundaries of the regions are most challenging to calculate with sufficient precision. Due to the instability of the NwAFC, combined with a relatively larger (monthly) period of data averaging and medium resolution of the available data, even a small change in the position of the transect can lead to a significant change in the integral flux through the section, e.g., from 10 to 75 TW for the Svinoy–Voring section. These uncertainties must be taken into account when calculating balances within the studied areas.



Ocean-atmosphere heat-fluxes were computed over the four areas: **A** (limited by the transects Svinoy and Jan–Mayen transects); **B** (between Jan-Mayen and Bear Island); **C** (between the transects Bear Island – Sorkapp); **D** (the transects Sorkapp and Fram strait).

## 2.6 Advective heat fluxes

Total oceanic heat flux through the transect (Q) is computed by integrating the heat flux values in the grid points eq. (1):

$$Q_i = \rho_0 C_p (T_i - T_{ref}) V_i dxdz ,\qquad\qquad(1)$$

where $\rho_0$ =1030 kg m$^{-3}$ is the mean sea water density; $C_p$ = 3900 J kg$^{-1}$°C$^{-1}$ is the specific heat of water; $T_i$ is the sea water temperature in each grid-point and on each depth level, $T_{ref}$ is the "reference temperature" of sea water, $V_i$ is the current speed module perpendicular to the transect, dx is the distance between the stations, dz is the thickness of the water layer.

When comparing the values of heat fluxes given by various authors (Skagseth et al., 2008; Smedsrud et al. 2013), it is necessary to take into account various choices of the "reference temperature". There is no justified algorithm for selection of the reference temperature (Schauer and Beszczynska-Möller, 2009). Here as the "reference temperature", we use $T_{ref}$ = 0 °C, as in most of the previous studies in the region (Hansen et al., 2008, Skagseth et al., 2008; Smedsrud et al. 2013, Bacon et al., 2015; Walczowski, 2014). Experiments show that modification of the absolute value of the heat fluxes and different choice of the reference temperature have only minor effects on the interannual variations, which are of primary interest here. In this study we integrate the advective heat fluxes over the Atlantic water layer (described in Table 1).

## 2.7 The Atlantic water in the eastern Nordic Seas

Although the term "Atlantic Water" is widely used in a number of studies, there is no common criterion for the definition of AW in the Nordic Seas. Depending on the goals and research areas, different criteria based on temperature, salinity, potential density and other parameters, as well as different threshold values were used (Table 1).

Table 1. Criteria used for definition of the AW in different studies

| Region | Criteria of the AW (T is temperature, S is salinity) | reference |
|---|---|---|
| Nordic Seas as a whole | T >1°C | K.A.Mork and J.Blindheim, 1999 |
| Norwegian Sea | T>5°C, S>35 | Orvik, K.A. et al., 2001. |
| Svinoy section | S>35 | K.A.Mork and Skagseth, Ø., 2010 |





| | | |
|---|---|---|
| Nordic Seas and Fram Strait | AW temperature = mean temperature in a layer of 70-500 meters, current direction to the north | Muilwijk, M. et al, 2018 |
| West-Spitsbergen current | T >2°C, S >34,88 | E.D.Cokelet et al, 2008 |
| Nordic Seas | current speed >30 cm/s | Orvik, K.A. and Niiler, P.P., 2001. |
| Arctic Ocean | T >0°C | Polyakov et al., 2017 |
| Southern boundary of Nordic Seas | T >5°C, S >35 | Wekerle et al., 2017 |
| Barents Sea opening | T >3°C, S >35 | |
| Southern boundary of Fram Strait | T >2°C, S >35 | |
| Southern boundary of Nordic Seas | T > 7 - 10,0 °C | Beszczynska-Möller, A., et al 2012 |
| Barents Sea opening | T > 6 – 6,5 °C | |
| Southern boundary of Fram Strait | T > 3 – 3,5 °C | |
| Svinoy section | T > 7-8°C, S >35,2 | Walczowski, W., 2014 |
| Gimsoy section | 6-7°C, S > 35,15 | |
| Fran Strait | T > 2°C, S > 34,95 | |
| From Svinoy section to 79°N | Potential density = 27,5 – 28 kg/m$^3$ | Furevik, T. et al., 2007* |
| Svinoy, Gimsoy, Sorkapp (layer 50-200 meters) | S = 34,9 – 35,3 <br> T= 2,5 – 9,5 °C | |

In this study, following Furevik et al. (2007), we limit the AW from below using the potential density threshold, which largely corresponds to the temperature and salinity thresholds, used in alternative studies. Furevik et al. (2007) give a rather broad

range of the threshold potential density values, increasing northwards. Due to the densification of the AW as it moves north (Latarius and Quadfasel, 2016), it is necessary to choose different parameters for different regions. To select the optimal density threshold values, the time-mean depths of various isopycnals from this range were overlaid on the vertical distribution of temperature and salinity at the transects across the Norwegian Atlantic Current (NwAC) (Fig. 4.). This allows testing the criteria against temperature and salinity thresholds used to define AW in different areas of the Nordic Seas in other studies

(Walczowski, 2014, Beszczynska-Möller et al, 2012, K.A. Mork and Skagseth, Ø., 2010). From our analysis, 27.8 isopycnal was selected for the Svinoy section, 27.85 for Voring, 27.9 for Jan Mayen, 27.95 for Bear Island, 28 for Sorkapp, Isfjord and the north of the Fram Strait. In between the sections, the isopycnals from the southern section were used.




Figure 4. Temperature (left column) and salinity (right column) along the selected transects from the south to the north. a, b, c, d – temperature on Svinoy, Jan Mayen, Sorkapp and Fram sections accordingly; e, f, g, h – salinity on Svinoy, Jan Mayen, Sorkapp and Fram sections accordingly. Isopycnals of potential density 27.8 (black), 27.85 (red), 27.9 (yellow), 27.95 (magenta), 28 (green) are overlaid, thicker dashed line highlights isopicnal outlining the AW layer.

## 2.8 Vertical mixing

Vertical turbulence heat flux through the base of the upper layer is estimated as:

$$Qvert = C_p \rho_0 K_z \Delta T / \Delta z , \qquad (2)$$



Where ΔT is the temperature differences between the lower boundary of AW and surrounding waters; $\Delta z = 100$ m. Two methods were used for obtaining $K_z$ values: (1) $K_z = \text{const} = 10^{-5}\,\text{m}^2\,\text{s}^{-1}$ (Fer et al., 2018) and (2) $K_z$ was estimated through the

Richardson number (Timmermann and Beckmann, 2004), which combines Pacanowski and Philander (1981) parameterisation with a diagnostic scheme using the Monin-Obukhov length.

In the second case (Pacanowski and Philander, 1981)), the diffusion coefficient is estimated as:

$$K_z = \frac{k^{uv}}{1+5Ri} + k_b^{ts} \,,\tag{3}$$

where

$$k^{uv} = k_{pp}^{uv} + k_{mo}^{uv} \,,\tag{4}$$

$$k_{pp}^{uv} = \frac{v_0}{(1+5Ri)^2} + k_b^{uv} \,,\tag{5}$$

$v_0 = 0.01\,\text{m}^2/\text{s}$, $k_b^{uv} = 10^{-4}\,\text{m}^2/\text{s}$, $k_b^{ts} = 10^{-5}\,\text{m}^2/\text{s}$.

The gravitational instability (convection) adds an additional term (Timmermann and Beckmann, 2004):

$$k_{mo}^{uv} = \begin{cases} 0.01\,\frac{m^2}{s}, & for\ |z| < \widehat{h}' \\ 0\,\frac{m^2}{s}, & for\ |z| \geq \widehat{h}' \end{cases} \,,\tag{6}$$

where $\widehat{h}'$ is the vertical scale of the length defined by the Monin-Obukhov length. In the case of our study no convective mixing through the lower interface was registered, so we set $k_{mo}^{uv} = 0$ .

Substituting all coefficients to the eqs. 3-5, we obtain:

$$Kz = \frac{k^{uv}}{1+5Ri} + 10^{-5} \,,\tag{7}$$

As currents in the ARMOR3D dataset are geostrophic, we calculate the Richardson number from the horizontal density
gradients using the geostrophic relations:

$$Ri = \frac{N^2}{(\frac{dU}{dz})^2 + (\frac{dV}{dz})^2} = N^2/R \,,\tag{11}$$

where $N^2 = \frac{g}{\rho_0} \times \frac{d\rho}{dz}$ is the buoyancy frequency, $R = \frac{g^2 ((\frac{d\rho}{dy})^2 + (\frac{d\rho}{dx})^2)}{\rho_0{}^2 f^2}$ and $f$ is the Coriolis parameter:

The results of $K_z$ estimation using the methods suggested in (Fer et al., 2018) and (Timmermann, R. and Beckmann, A., 2004) were found to be similar.

## 3. Results and discussion

### 3.1 Validation of ARMOR3D heat fluxes

To validate the ARMOR3D estimates, we compare the statistical properties of all available mooring observations in the Fram Strait (AWI F1-F10) with those in the nearest grid-point of the ARMOR3D dataset. To obtain more homogeneous data series suitable for further comparison, a preliminary filtering is applied to the moorings data. The processing steps include removal





of outliers and monthly averaging of the data to cope with the ARMOR3D temporal resolution. Then the ARMOR3D and the moorings data are binned to 100 m vertical bins, centered on the position of the available moored instruments. Figure 5 shows an example of statistical comparison of the time series of temperature, velocity and heat fluxes for the two datasets at mooring F5 (located in the eastern Fram strait 78,5° N and 6° E). The Taylor diagrams (Taylor, 2001) show that the temperature variability is well reproduced by the ARMOR3D dataset (the correlation coefficient is 0.7). On the other hand, current velocity

(and the heat flux) derived from ARMOR3D, shows lower interannual and seasonal absolute values, as well as variability, compared to in situ data. However, the seasonal pattern of the heat fluxes, as well as the interannual one, are reproduced in ARMOR3D with reasonable accuracy (the correlation coefficient is 0.6). The meridional velocity component is much better reproduced by ARMOR3D, compared to the zonal one, which is because the main geostrophic flow in the region is directed northward. Therefore, we may expect a higher accuracy of the heat fluxes across the zonal sections, compared to the near-

meridional ones.

In the areas with the presence of the drifting ice (the East Greenland Current) and at deeper water levels, the performance of ARMOR3D in comparison to the mooring data naturally decreases. This is due to a decrease in the accuracy of the satellite altimetry in areas with sea ice and accumulation of the errors while integrating the density gradient downwards. For the present study, focused on the upper 500-meter layer and in the regions with no winter ice cover, we consider the results from the

ARMOR3D dataset reasonably well representing the interannual variability of the heat fluxes.







Figure 5. Validation of ARMOR-3D (blue) against in situ data at mooring F5 (red) located in the WSC at **78,5° N 6° E**: a –
water temperature (ºC), b – zonal current velocity U (cm s⁻¹) and c – meridional current velocity V (cm s⁻¹). Left -Taylor
diagrams (ARMOR-3D is point B, in situ – point A), center – data time series, right - seasonal cycles. Data are averaged in
50-150 m layer.


### 3.2 Temporal variability of heat fluxes along the NwAC

Svinoy section is one of the main sites where AW inflow into the Nordic Seas is monitored continuously (Orvik and Niiler,
2002; Raj et al., 2018). The heat fluxes are calculated over AW layer limited from below by the isopycnals presented in Section
2.7. The heat advection across the section is split between three main cores of the AW: the coastal branch at 10° E (NwACC),

the slope branch between 5 and 6° E (NwASC) and the polar frontal branch between 2 and 3° E (NwAFC). Our analysis shows





that the largest heat flux is directed northward along with the NwASC. From the Svinoy (406 TW) to the Jan Mayen sections (341 TW) the heat advection decreases by about 1/3. This is consistent with the observed significant heat loss of NwAC to the interior of the Lofoten Basin, the main heat reservoir in the Nordic seas (Bjork, 2001; Bosse et al., 2018). Mean integrated heat fluxes have more or less similar seasonal patterns at all transects and for all years: the heat flux decreases in summer and

increases in winter. The seasonal cycle is regulated by the seasonal variability of the current velocity, which is higher in winter. This is in line with the previous results on the heat transport in the area (Skagseth et al., 2004, 2008; Mork and Skagseth, 2010). The winter maximum of the NwASC is explained by a higher sea-level gradient caused by an increased Ekman pumping associated with stronger northerly winds along the Scandinavian coast (Skagseth et al., 2008; Mork and Skagseth, 2010). Further in this paper we extend the analysis by Chafik et al. (2016), who studied the consistency of interannual variations of

NwASC along the Voring plateau, analyzing the interannual variations of heat fluxes further north up to the northern part of the Fram Strait.

On average over the study period (1993–2017), the major heat flux, which enters the Norwegian Sea, passes north with the NwAC across the Svinoy section 1 (406 TW). Another 132 TW enter the study region from the west, across a deep Aegir Ridge and the Jan Mayen Fracture Zone (western boundary of region A shown in Fig. 6). These western boundary ridges (as

well as Mohn-Knipovich ridges further north) do not rise over 1500-2000 m depth, far below the lower AW limit. The dynamic boundary of the bottom trapped NwAFC, limiting the study region from the west, is a subject of a relatively intensive cross-frontal exchange (Raj et al., 2019). Half of the overall incoming heat passes further north through the Jan Mayen Section (~341 TW), the northern boundary of region A. Then about 89 TW enter the Barents Sea through the Barents Sea Opening and 131 TW continue north across the Bear Island Section (northern boundary of region B). Only 54 TW, around 1/10 of the heat

entering the Lofoten Basin, reach the Fram Section (northern boundary of region D). The heat fluxes across the western boundary of the regions B–D are negligibly small and highly variable. In region D, the westwards heat flux of about 3 TW represents the recirculation of AW southwestwards to the Greenland Sea. These estimates should be treated with caution as the values depend on the position of the transects and rather small changes in it affect the resulting fluxes. A significant amount of heat is lost due to the vertical mixing across the AW boundary (Fig. 6), in particular in the Lofoten Basin, where convection

across the AW lower boundary is episodically observed (Bosse et al., 2018; Fedorov et al., 2019). The main components of the heat balance of the regions A–D are schematically shown in Fig. 6. The imbalances obtained account to 10-20 % of the incoming heat fluxes. These reflect the warming of the AW in the Norwegian Sea. We also should take into account uncertainties of the estimated oceanic heat advection. Due to the uncertainty in the reference temperature, Schauer and Beszczynska-Möller (2009) suggest treating the oceanic heat fluxes in terms of their variability, rather than relying on their

absolute values.





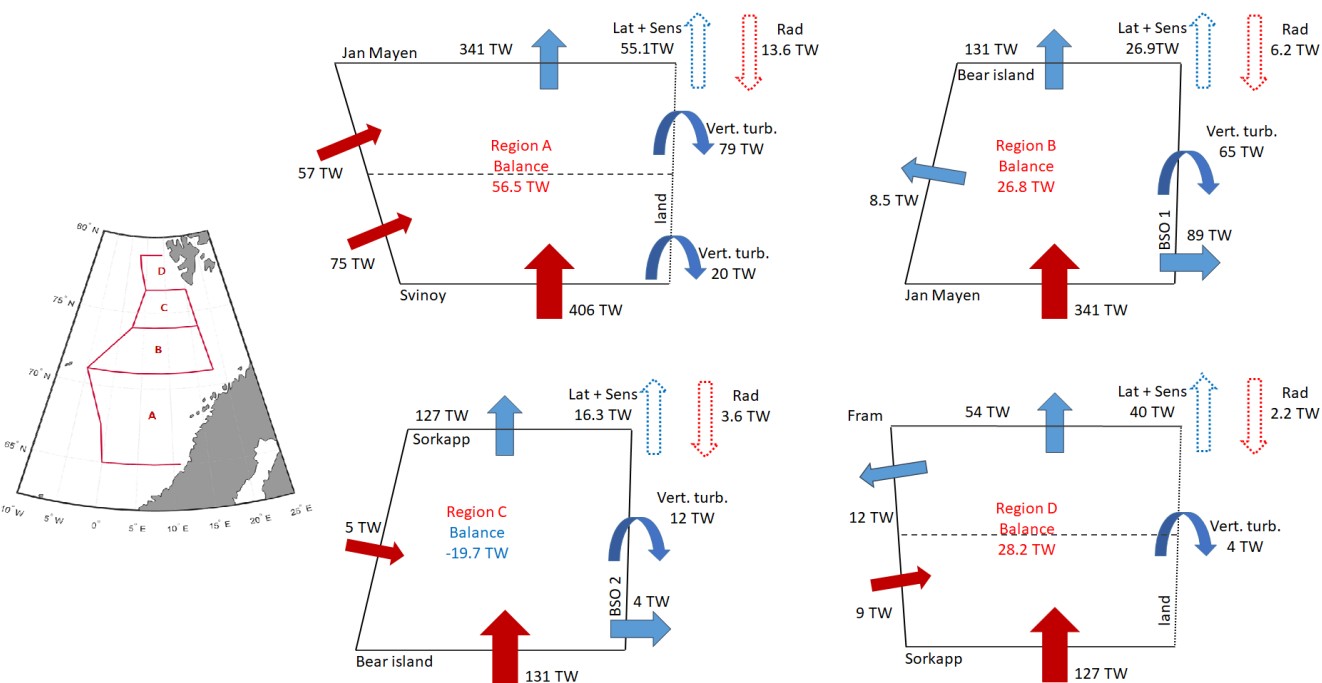

Figure 6. Components of the heat balance for the Regions A–D. Straight red (blue) arrows represent the oceanic heat fluxes
entering (leaving) the study regions, dashed arrows represent latent and sensible heat fluxes from ocean to atmosphere (blue)
and radiation from atmosphere to ocean (red), curved arrow represents vertical turbulent heat flux.

On its way north AW changes its properties through mixing with the surrounding water and the ocean-atmosphere exchange.
Propagating at different rates, which vary in time, these multiple transformations contribute to the loss of correlations between
heat fluxes across the transects (Fig. 7). Skagseth et al. (2008) found a certain coherence between the temperature and salinity
variations at the Svinoy and the Sorkapp/BSO sections on decadal time scales. Our results show that there is a significant loss
of correlations along the AW pathway, dropping to insignificant levels north of the Sorkapp section. The strongest loss of
correlation is found between Svinoy and Jan Mayen sections, as well as between Sorkapp and Isfjord. The loss of a consistent
interannual variability between Svinoy and Jan Mayen sections along the NwAC can be explained by high activity of oceanic
eddies which redistribute the heat over the Lofoten basin (Dugstad et al., 2019; Raj and Halo, 2016). The same explains the
correlation loss between Isfjord and Sorkapp sections (von Appen et al., 2015). A drop in correlation value may result from
1–1.5 year period, which is the time required for an anomaly to propagate from 63 to 76° N, given the mean anomaly
propagation velocity of 3 cm s$^{-1}$ (Walkzowski, 2014). However, the cross-correlation analysis suggests the maximum
correlations at zero time lag, which suggests rather simultaneous forcing at all the sections.
After removing the trend in the deseasoned data, the cross-correlation between the sets of the southern sections (Svinoy to Jan
Mayen) and the northern sections (Sorkapp to Fram) increases, while this procedure does not affect the correlations between





Svinoy and Jan Mayen (Figure 7b). This indicates the different signs of the long-term variability in the southern and the northern sections (also present in Figure 9).

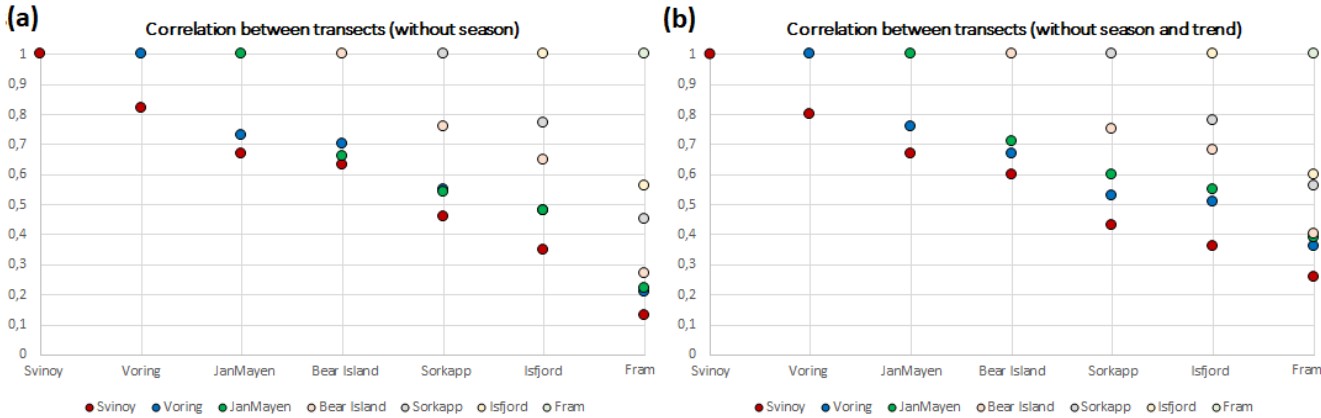

Figure 7. Correlation coefficients between monthly mean values of the heat flux across the marked transects a - seasonal cycle is removed, b – seasonal cycle and trend are removed


The results above suggest different mechanisms controlling heat transport at the southern and northern sections along the NwASC. Skagseth et al. (2008) and Raj et al. (2018) suggest NAO to be the principal agent in modulating the AW transport along the southwestern Scandinavian coast. On the other hand, Lien et al. (2013) have shown that the relative strengths of the branches of AW along the western Spitsbergen and in the Barents Sea are strongly affected by a regional atmospheric

circulation pattern over the Spitsbergen and the northwestern Barents Sea: a higher AW transport in the Barents Sea is accompanied by a lower transport through the Fram Strait. Hence, we hypothesize that NAO is more affecting the southern part of the AW pathway, while the local atmospheric circulation pattern in the Nordic Seas impacts the northern part of the AW pathway. Only the heat fluxes across the southern sections show significant moderate positive correlations with the NAO, AO and EA indices $(0.34 - 0.47)$ (Table 2). Previously, Chafik et al. (2015) showed that NAO is not the driving mode for the

AW influx through the Fram Strait and that the regional atmospheric circulation is the main driving factor. For the northern sections the correlations go to zero.

A consistent sign of the correlations is obtained between the advective heat fluxes at all sections and the weather types C and W (Table 2), although only the correlations with the heat fluxes across the southernmost and the northernmost sections are significant. Even though not always significant, the correlation coefficients are always positive with the western weather type

W and are always negative with the central weather type C. This suggests a possible existence of the large-scale forcing pattern, responsible for the in-phase variations along the NwAC. With the weather type W, the winds are intensified along the Scandinavian coast accumulating the water along the coast and intensifying the NwASC (Fig. 8a). The intensification of the northern NwAFC may result from a stronger gradient of the wind speed west of Spitsbergen, which provides the sea-level drop across the NwAFC due to the divergence of the Ekman fluxes. With the weather type C, the main winds are directed towards





the Scandinavian coast, thus the Ekman transport is directed south which decreases the NwASC (no water accumulation along the Scandinavian coast), whereas the weak gradients of the wind speed west of Spitsbergen reduce the Ekman divergence which is not favorable for the intensification of the NwAFC (Fig. 8c). The lack of correlation with type E can be explained by the overall weaker winds over in the eastern Nordic Seas (Fig. 8b).

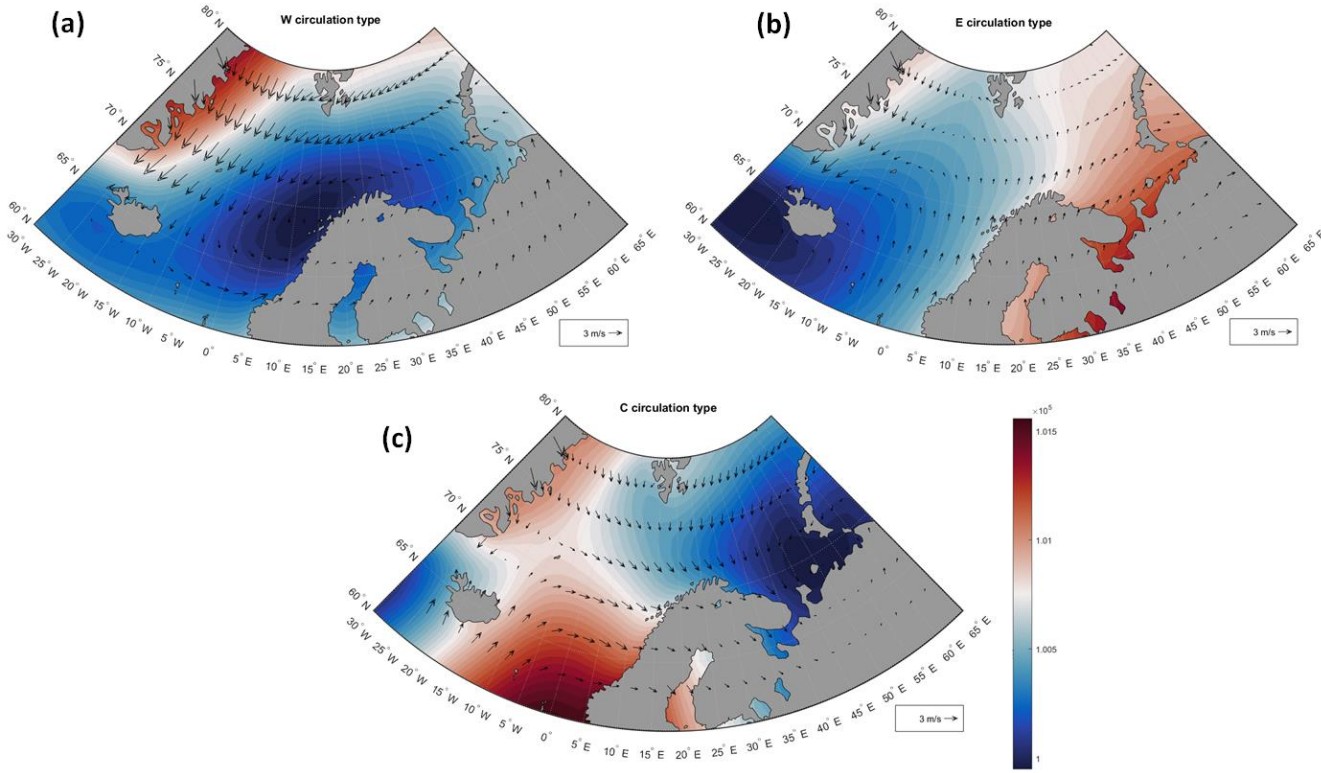


Figure 8. Mean sea level pressure fields (shaded, Pa) and dominant wind patterns (vectors) over the North Atlantic associated with circulation types: a - W, b - E and c - C

345                  Table 2. Correlation coefficients between heat fluxes and indices (bold italic – significant values)

|  | Svinoy | Voring | JanMayen | Bear Island | Sorkapp | Fram |
|---|---|---|---|---|---|---|
| **NAO** | *0.46* | *0.45* | 0.26 | 0.09 | 0.06 | -0.02 |
| **AO** | *0.47* | *0.45* | 0.23 | 0.18 | 0.17 | 0.05 |
| **EA** | *0.39* | *0.34* | 0.18 | 0.13 | -0.15 | -0.26 |
| **AMO** | 0.13 | 0.23 | *0.33* | 0.21 | 0.09 | 0.06 |
| **E** | -0.09 | -0.02 | -0.06 | 0.03 | -0.14 | -0.25 |



| C | *-0.46* | *-0.41* | -0.29 | -0.26 | *-0.33* | -0.24 |
| W | *0.39* | *0.30* | 0.25 | 0.15 | *0.35* | *0.38* |

From 1993 to 2017 there are no pronounced positive or negative long-term trends (Fig. 9 a,b) in the heat flux across most of the sections (apart from the BSO section, where the consistent tendency of the heat flux to grow is observed – not shown, see Kalavichchi, Bashmachnikov, 2019). At Svinoy section there is a certain tendency of oceanic heat advection to grow, the most
pronounced since 2010. This is in line with the recent warming of the AW after 2010–2011 in the Norwegian Sea derived from the Argo float profiles (Mork et al., 2019). However, at the Fram section, the heat advection increases only in the beginning of the 2000s, and since 2005 it started to decrease. Thus, despite the general increase in the water temperature in the south of the region, the northern section does not demonstrate a positive trend in the heat fluxes during the latest decades. This is one of the factors reducing the correlations.
To detect the hidden periodicities in the heat fluxes, the wavelet analysis with the Morlet mother wavelets is applied (Torrence and Compo, 1998). In all the transects we distinguish the main periodicities of 3 years and of 5–6 years (Fig. 9c,d). The wavelet amplitudes decrease northwards, along with the decrease of the mean heat fluxes. The cross-wavelet diagram shows a high coherence of heat fluxes in the Svinoy and Fram Strait sections at time-periods of 2–5 years, the variability at these periods occurs in phase. This suggests that on intra-decadal time scales there is a certain coherence in the oceanic heat advection along
the NwAC. W and C indexes have similar variability with the time scales of 2–3 and 5–7 years, which further supports the existence of the link between the oceanic heat advection along the NwAC and the W–C weather patterns.







Figure 9. Time series (a, b) and wavelet diagrams (c, d) of interannual variations of heat fluxes: on the left – Svinoy section, on the right – Fram section, e – cross-wavelet diagram between the Svinoy and Fram sections.


## Conclusions

The present analysis suggests a certain consistency of the heat fluxes along the path of the NwAC through the Nordic Seas. This consistency results from the high cross-wavelet coherence between the heat fluxes at the southern and the northern





sections at some interannual time scales. This coherence results from 3 and 5–6 year oscillations dominating the short-term
interannual variability. Wind patterns corresponding to the C and W weather types may serve the forcing mechanism,
increasing/decreasing the heat advection along the whole path of the NwAC.

However, there are notable differences in the heat fluxes which result in a decrease of the correlations. Particularly strong
drops of correlations are observed across the Lofoten Basin (between the Svinoy and Jan Mayen sections) and north of the
Bear Island. The reasons are the opposite tendencies in the long-term variability (after the mid 2000s) and differential damping
of the detected oscillations (the longer oscillations are damped more effectively while progressing north: the amplitude of the
5 years oscillation drops by 50% from Svinoy to the Bear Island and further on by 60% from the Bear Island to the Fram
section, while the amplitude of the 3 years oscillation drops by 40% progressing from Svinoy to the Bear Island and further on
only by 8% from the Bear Island to the Fram section). One of the reasons for this behavior may be the dependence of the heat
fluxes across the southern sections on NAO type patterns, while it practically does not influence the northern sections. In turn,
the northern sections depend on the variability of the local cyclonic wind patterns, centered in the north-western Barents Sea
(Lien et al., 2013; Chafik et al., 2015). The observed variability of the heat fluxes is mostly shaped by the variations in the
current velocity and is only marginally influenced by the changes in temperature of the AW.

The oceanic heat inflow in the regions (A–D) is largely balanced by the heat release to the atmosphere and by vertical mixing.
The first dominates in the northern part of the study region (west of Spitsbergen), while the second – in its southern part (the
Lofoten Basin). The imbalances form from 10 to 20% of the incoming heat and encompass the heat fluxes by the mesoscale
eddies (Raj et al., 2020; Bashmachnikov et al, 2020). The imbalances lead to the observed warming of the eastern Nordic Seas.
Discussing the balances, the errors in the heat fluxes should be taken into consideration. In particular, the positions of the
transects highly affect the results. However, these errors in the absolute values of the fluxes practically do not affect the
interannual variability discussed above.


**Author contribution:**

Anna Vesman – Conceptualization, Data curation, Formal analysis, Investigation, Validation, Visualization, Writing – original
draft preparation; Igor Bashmachnikov - Conceptualization, Funding acquisition, Investigation, Project administration,
Supervision, Writing – review & editing; Pavel Golubkin - Data curation, Formal analysis (atmospheric heat fluxes), Writing
– review & editing; Roshin Raj - Conceptualization, Funding acquisition, Writing – review & editing

**Funding:** The research was supported by the Russian Science Foundation (RSF), project No 18-17-00027 and by the
Norwegian Research council funded ARCNOR project (No. 261743).



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
