# Peer review of "The coherence of the oceanic heat transport through the Nordic seas: oceanic heat budget and interannual variability"

_Ocean Science, 2020_

## Referee Comment (RC1) · Anonymous Referee #1 · 2 Jan 2021

Review on "The coherence of the oceanic heat transport through the Nordic Seas: oceanic heat budget and interannual variability" by Vesman et al.

**General comments**

Using observation-based datasets, Vesman et al. studied the connectivity of advective heat flux across a number of sections in the Norwegian Sea. They have further discussed the driving mechanisms for the heat flux variability, including NAO, AO, the meridional (C) and western (W) weather types. Results from this study have implications on the heat (and salt) transported to the Nordic and Artic Seas, which is important to understand the high latitude climate state and variability.

The paper is overall clearly written and the focus on the heat flux connectivity is of interest to the community. However, throughout the paper, the authors computed heat flux with a reference temperature along a non-conserved section. This is actually a calculation of temperature flux instead of heat flux, and the difference between the two can be huge (see Forget and Ferreira, 2019). While the variability may not be significantly influenced, as the authors have suggested, the mean heat budget discussed in section 3.2 (Figure 6) is meaningless. I strongly suggest the authors to carefully address this issue before considering publication. One possible approach is to calculate heat flux along closed section and apply mass conservation. Another is to repeat calculations with different reference temperatures to test the sensitivity of the results.

**Detailed comments**

[1]. Line 23: This sentence is hard to understand without reading the manuscript. Suggest to re-write.

[2]. Line 48-49: For those without a knowledge on the current system (e.g. Yermak brach) or topography features (Yermak Plateau) in the Norwegian Sea, it is very difficult to navigate. I suggest labeling them in Figure 1.

[3]. Line 67: Where is the Norwegian Atlantic Coastal Current? Could you also label it in Figure 1?

[4]. Line 87: I suggest adding a paragraph describing motivations of this work.

[5]. Section 2.3: Estimates based on different atmospheric products could be quite different (Chavik and Rossby, 2019). I strongly suggest estimating with different products to derive an ensemble mean and a standard error.

[6]. Section 2.5: To derive heat flux, it is necessary to have a closed basin (from coast to coast). While I understand the authors' focus is on heat carried by AW, such a calculation is only a temperature flux and should not be used to infer heat changes (heat changes are not only influenced by warm waters flowing northward but also by cold waters flowing southward at the section).

[7]. Section 2.8: Vertical mixing or diapycnal mixing? Are you estimating the mixing at the base of the AW, which is along an isopycnal? If so, how robust is it to use vertical mixing coefficient $K_z$ to estimate mixing across an isopycnal?

[8]. Line 238-239: The correlation seems to result from the trend. What is the correlation after detrending the time series?

[9]. Figure 5: Suggesting adding error bars to the time series plot.

[10]. Figure 6: Suggest adding uncertainties to the budget analysis. There is a clear difference of the mean meridional velocity between the ARMOR and the mooring (Figure 5), implying large uncertainties in the estimated advective heat flux. Again, the calculated heat flux is really a temperature flux, whose mean may be significantly modified with a different reference temperature.

[11]. Line 301: The correlation between Svinoy and Jan Mayen is as high as 0.7 according to Figure 7. Why is that a loss of correlation?

[12]. Figure 9c: There seems to be an increase of the dominant period with time. For example, in months 192-288, there is a dominant period of >84 months. Is there an explanation for that?

**Reference**

Forget, G. and Ferreira, D., 2019. Global ocean heat transport dominated by heat export from the tropical Pacific. Nature Geoscience, 12(5), pp.351-354.

Chafik, L. and Rossby, T., 2019. Volume, heat, and freshwater divergences in the subpolar North Atlantic suggest the Nordic Seas as key to the state of the meridional overturning circulation. Geophysical Research Letters, 46(9), pp.4799-4808.

---

## Referee Comment (RC2) · Anonymous Referee #2 · 5 Jan 2021

Review on "The coherence of the oceanic heat transport through the Nordic Seas: oceanic heat budget and interannual variability" by Anna Vesman, Igor Bashmachnikov, Pavel Golubkin, Roshin Raj submitted to the Ocean Science journal.

**General comments**

The authors investigated the heat flux in several sections northward in the Nordic Seas along the pathways of Atlantic Water into the Arctic. This was done using the ARMOR3D dataset that derives from in-situ temperature and salinity data and satellite data. The heat flux variability is discussed in relation to atmospheric forcing that includes different weather regimes and North Atlantic Oscillation. The topic is of high scientific interest and this work could contribute to our understanding of the heat transport toward the Arctic. However, it is not clear what is the new findings in this paper. This should be more highlighted and discussed in the paper. The paper is well structured but the discussion should be improved (as mentioned above) and there are also too many errors or unexplained (or not shown) statements in the paper (see my comments below) to be accepted for publication.

**Detailed comments**

Since the ARMOR3D and mooring observation have monthly values and there is strong seasonality in both data sets, there is not unexpected that the correlation between them is relatively high (Line 231-245 and figure 5). To compare the two datasets for interannually variability the seasonal signal should be removed before the correlation analysis.

Line 56-58: Several currents are mentioned but the East Iceland Current transports Arctic water and what is the West Icelandic Current? the (North-Icelandic) Irminger Current seems to be more relevant. Also, what is the Norwegian Atlantic Coastal Current? The location of the most central currents should be included in the map (Fig. 1.).

Line 90: What stands ARMOR3D for? There are many abbreviations without explanation (e.g. ARMOR3D, CMEMS, NACLIM, CLIMODE, COARE, …).

Line 119: "Vangengeim suggested…". Reference is missing.

Line 130: "Vangengeim – Giers classification…" What is this classification?

Line 151: The depth interval and the reference temperature for the heat fluxes are missing.

Line 160-162: "…even a small change in the position of the transect can lead to a significant change in the integral flux through the section…. These uncertainties must be taken into account when calculating balances within the studied areas" I cannot see that this has been done or have been discussed.

Line 188-189: The authors use potential density thresholds for the definition of AW. Table 2 seems to be unnecessary and has several errors: Mork and Blindheim (1999) does not exist, Orvik et al. 2001 only studied the Svinøy section and not the whole Norwegian Sea (the ref.

is also missing), Orvik and Niiler (2001) used 30 cm/s currents as AW pathways (not as definition of AW), reference of Furevik et al (2007) is missing, etc.

Line 205-208: The reason for why dz=100 m was chosen is lacking? "dT is the temperature differences between the lower boundary of AW and surrounding waters" Is the surrounding waters 100 m (=dz) below the boundary of AW?

Line 209-229: I don't see that this calculation is necessary. Why not just use the constant value from Fer et al. with the reference. The uncertainties of Kz is probably less important than compared to other variables (e.g., dT and dz) or the chose of the reference temperature in eq. 1.

Line 286-288: "The imbalance account to 10-20% …that reflect the warming of AW…We should take into account uncertainties…" I cannot see that the uncertainties are discussed in the paper (e.g., what will be the uncertainties), and if this imbalance reflects the warming, how much will this be (in temperature/heat). This should then be compared with other works.

Line 300: "…dropping to insignificant levels…". What is the significant level?

Line 301-303: That the correlation drops between Svinøy and Jan Mayen sections might also be due to the AW lies deeper and has longer residence time in the Lofoten Basin (e.g., Nilsen and Falck, 2006; Skagseth and Mork, 2012).

Line 306-307: "…the cross-correlation analysis suggests the maximum correlations at zero time lag". Is this done?

Line 381-382: "… is mostly shaped by the variations in the current velocity…" This should then be shown.

Line 385-387: Could the 10-20% of the incoming heat be due to reduced air-sea heat loss? See also my comments on Line 286-288.

Figure 1: the position of NwAFC is wrong.

Figure 2: Are the currents from ARMOR3D? At which depth? Is it same as for temperature (50m)?

Figure 7. The correlation should also be done with time lags. While the velocity might have no or short time lag the temperature might have 1-2 year time lag (see e.g., Holliday et al., 2008; Skagseth et al., 2008; Chafik et al., 2015) which was also mentioned by the authors.

Figure 9. The figure with wavelet amplitudes seems to be wrong. The amplitude should only be positive. Why is the heat flux integrated to 500 m? Why not use the boundary level of AW defined in the paper?

Table 2. What is the significant level.

---

## Referee Comment (RC3) · Anonymous Referee #3 · 6 Jan 2021

Review of the manuscript "The coherence of the oceanic heat transport through the Nordic seas: oceanic heat budget and interannual variability" by Anna V. Vesman and co-authors, submitted to Ocean Science.

**General comments**

Vesman and co-authors seek to investigate the variability in transport of heat through the Nordic seas towards the Arctic Ocean using a gridded dataset of monthly temperature, salinity and velocity fields derived from in situ and satellite data, and to frame this variability in terms of atmospheric forcing of the ocean. An improved understanding of heat transport in this region would be valuable, given its significance for changing ice cover in the Arctic in the coming years.

The paper is clearly structured, and well written. It is, however, difficult to assess the significance of the results presented here in the absence of any discussion of the errors associated with them, which the authors acknowledge are likely to be significant. The authors have attempted to provide some explanation of the physical basis behind the correlations in variability in heat fluxes that they see at various locations along the path of Atlantic Water (AW), but unfortunately this is unconvincing because the patterns of atmospheric forcing in terms of weather types that they present are inconsistent with their results. I offer some more detailed comments below, but I suggest that these shortcomings need to be addressed before the manuscript can be considered for publication.

**Specific comments**

1.   Line 123: I am not familiar with this particular categorisation. Can you provide some background – how it is derived, and why it is an appropriate description of the atmospheric patterns seen over the Nordic seas – for the benefit of readers who are unable to follow the Russian-language references?

2.   Figure 2: The maps are small and the detail difficult to make out, but it looks as if you might be losing some of the northward AW flow and periodic southward recirculation at the eastern end of some of your transects, particularly at the southern end of the Barents Sea Opening. The current is strong here, and its position varies a fair amount. (See, for example, Wang et al. 2019.) You mention at Line 160 that your results are sensitive to the position of the transects in relation to the western boundaries of your regions, but do not, so far as I can see, provide any quantification of the uncertainties associated with choice of position.

3.   Line 204: "the base of the upper layer". Is this the AW layer?

4.   Line 232: "we compare the statistical properties of all available mooring observations….with those in the nearest grid-point of the ARMOR3D dataset." Would you see a better correlation if you compared the mooring observations with interpolated values from the ARMOR3D dataset? LaCasce 2005 found low spatial correlations between current meter readings taken from moorings only a few kilometres apart on the Norwegian Slope, and similar lack of correlation might be expected between current meter observations and ARMOR3D grid points over a similar distance. Nevertheless, the spatial resolution of satellite observations underlying the gridded dataset might be insufficient for further interpolation to offer an improvement.

5.   Line 236: "data are binned to 100 m vertical bins". Why 100 m?

6.   Line 240: "current velocity...derived from ARMOR3D shows lower…..variability, compared to in situ data". Variability will depend on the scale over which values are averaged. The mooring data are collected at fixed points, so one might expect them to exhibit higher variability.

7. Line 259: The NwACC carries fresher water of Baltic origin, not AW. (Skagseth et al. 2008, in your reference list.)

8. Lines 261/2: Are these long term mean heat fluxes?

9. Figure 6: Balances can only be given to the precision of the least precise of the inputs, not to 0.1 of a terawatt. But more seriously, what are the error estimates for these calculations?

10. Figure 8. Are these mean sea level pressure fields, as the caption says, or anomalies? (The values on the colour bar are too small to be legible.) If you wish to relate variability in ocean transports to variability in atmospheric forcing, do you not wish to look at the anomalies from long term mean?

11. Lines 331-3: This doesn't seem quite right. The colour shading in Figure 8a. obscures the wind vectors, but they appear to point slightly northwards along the coast at Svinøy, but offshore at Spitsbergen. So we might expect to see some build up of sea level against the coast and consequent enhancement of the shelf current at Svinøy, but no similar Ekman effect at the more northerly transects.

12. Lines 334-7: It is a convergence or divergence of Ekman transport in association with the coast that generates the sea level gradients which lead to the variability in the geostrophic slope current. Ekman transport in a southerly (along slope) direction cannot, therefore, decrease the NwASC, because it does not involve convergence or divergence of transport. It should have no effect.

13. Line 337: The along slope winds shown for Type E in Figure 8b. do not appear to be significantly weaker than those for Type W in the same region.

14. Line 352: "since 2005 it [the heat advection] started to decrease". Figure 9b. appears to show a recovery of heat flux in the final two years. Do we not just we decadal-scale variability here, rather than any trend?

15. Line 358: How sensitive are these results to choice of transect? Would they show similar periodicity and coherence if you chose the Vøring and Isfjord transects, for example?

**Technical corrections**

All figures: small text is difficult to read. Can you make the labelling clearer? Colours are also difficult to distinguish in Figure 7.

Line 141: "the currents are strongly bottom trapped". Do you mean "topographically steered"?

Line 268: "stronger northerly winds". This English expression is commonly understood to refer to winds blowing from the north, whereas I think you are talking about winds blowing from south to north. I suggest "northward-blowing winds".

Line 272: Delete the comma after "heat flux". It changes the meaning of the phrase.

Line 320: "into the Barents Sea"?

Line 326: "the correlations go to zero". Go to zero, or just become small?

**References**

LaCasce, J. H. (2005). Statistics of low frequency currents over the western Norwegian shelf and slope I: current meters. Ocean Dynamics, 55: 213–221.

Wang, Q., Wang, X., Wekerle, C., Danilov, S., Jung, T., Koldunov, N., et al. (2019). Ocean heat transport into the Barents Sea: Distinct controls on the upward trend and interannual variability. Geophysical Research Letters, 46, 13,180–13,190.

---

## Author Comment (AC1) · 5 Mar 2021

General comments

Using observation-based datasets, Vesman et al. studied the connectivity of advective heat flux across a number of sections in the Norwegian Sea. They have further discussed the driving mechanisms for the heat flux variability, including NAO, AO, the meridional (C) and western (W) weather types. Results from this study have implications on the heat (and salt) transported to the Nordic and Artic Seas, which is important to understand the high latitude climate state and variability. The paper is overall clearly written and the focus on the heat flux connectivity is of interest to the community. However, throughout the paper, the authors computed heat flux with a reference temperature along a non-conserved section. This is actually a calculation of temperature flux instead of heat flux, and the difference between the two can be huge (see Forget and Ferreira, 2019). While the variability may not be significantly influenced, as the authors have suggested, the mean heat budget discussed in section 3.2 (Figure 6) is meaningless. I strongly suggest the authors to carefully address this issue before considering publication. One possible approach is to calculate heat flux along closed section and apply mass conservation. Another is to repeat calculations with different reference temperatures to test the sensitivity of the results.

Response: Thank you for your comments and suggestions, they helped to improve the quality of our work. Our study is focused on several closed upper ocean areas, limited from below by an isopycnal. The fluxes are estimated across all vertical sections, across the sea-surface and across the bottom isopycnal, thus closing the volumes studied. All water (warm or cold) passing through the boundaries of the selected water volume is taken into account. We also stress that the main goal of these computations was not estimating the heat balances over the selected areas (in fact, the tendencies in water temperature suggest misbalances), but deriving the main sources of the AW heat loss along its travel north. More detailed answers and corrections are provided below:

Detailed comments

[1]. Line 23: This sentence is hard to understand without reading the manuscript. Suggest to rewrite.

Response: Sentence was changed to: Line 20: "This is a result of different tendencies over the latest decades in the southern and the northern parts of the study region, as well as of a differential damping of the observed periodicities along the Atlantic Water path on its way north (the amplitude of 5–6 year oscillations drops significantly faster than that of 2–3 year oscillations)."

[2]. Line 48-49: For those without a knowledge on the current system (e.g. Yermak brach) or topography features (Yermak Plateau) in the Norwegian Sea, it is very difficult to navigate. I suggest labeling them in Figure 1.

Response: Authors agree that the map lacked important information and had some errors in the current directions. The map was modified and more information was added. Now Figure 1 includes: NwAFC - Norwegian Atlantic Front Current, NwCC – Norwegian Coastal Current, NCC – North Cape Current, WSC – West Spitsbergen Current, SB – Svalbard branch, YB – Yermak branch, EGC – East Greenland Current, EIC – East Icelandic Current, NIIC – North Icelandic Irminger Current, SIC – South Icelandic Current, FC – Faroe Current; VP – Voring plateau, LB – Lofoten basin, Sb – Spitsbergen, YP – Yermak plateau

Figure 1.Schematic map of oceanic circulation in the study region

[3]. Line 67: Where is the Norwegian Atlantic Coastal Current? Could you also label it in Figure 1?

Response:

Line 69: The phrase was changed to "The AW further enters the Barents Sea along the northern shelf of Scandinavia as the North Cape and the Norwegian Coastal (Murmansk) currents . . .." Norwegian Coastal Current was added to the map, previously erroneously named as the Norwegian Atlantic Coastal Current.

[4]. Line 87: I suggest adding a paragraph describing motivations of this work.

Response:

Line 90: The sentence was added: "In this paper we analyze the space-time variability in the advective heat fluxes along the AW pathways into the Arctic. The main motivation of this work was to understand to what extent the anomalies of the oceanic heat flux, entering the Nordic Seas from the south, are conducted into the Arctic. This shows whether the observations at the southern transects (i.e. Svinoy) are representative for

evaluation of the variability of the ocean heat advection to the Arctic on the interannual time scales. We also aim to investigating mechanisms behind a possible coherence loss along the AW path."

[5]. Section 2.3: Estimates based on different atmospheric products could be quite different (Chavik and Rossby, 2019). I strongly suggest estimating with different products to derive an ensemble mean and a standard error.

Response:

Comparison between different reanalyzes models show very similar interannual variability. Example of heat fluxes, averaged over region, is presented in Supplementary material, Figure S2.

Figure S2. Ocean atmosphere heat fluxes for region A, calculated using different datasets

[6]. Section 2.5: To derive heat flux, it is necessary to have a closed basin (from coast to coast). While I understand the authors' focus is on heat carried by AW, such a calculation is only a temperature flux and should not be used to infer heat changes (heat changes are not only influenced by warm waters flowing northward but also by cold waters flowing southward at the section).

Response:

It is true that this work doesn't take a basin coast to coast, as, in this study, we are interested only in the northwards path of Atlantic water. Our study is focused on several closed upper ocean areas, limited from below by an isopycnal. The fluxes are estimated across all vertical sections, across the sea-surface and across the bottom isopycnal, thus closing the volumes studied. All water (warm or cold) passing through the boundaries of the selected water volume is taken into account. We also stress that the main goal of these computations was not estimating the heat balances over the selected areas (in fact, the tendencies in water temperature suggest misbalances),

but deriving the main sources of the AW heat loss along its travel north. To further illustrate this complex problem of obtaining the heat balance, time series of the heat fluxes for different positions of the boundary transects, as well as for different reference temperatures were added to the Supplementary material. Important is that, with a single exception, the variations do not change the tendencies and the main patterns of the interannual variability of the heat fluxes. This suggest the robustness of the results obtained.

Lines 170-176 were changed to "The instability of the NwAFC, a relatively large (monthly) period of data averaging, the medium resolution of the available data, and anaccounted ageostrophic component can lead to a significant change in the integral flux through the section even with a relatively small change in the position of the transects. These uncertainties are taken into account when discussing the values of the ocean heat convergence in the subregions, limited by the transects: A (limited by the transects Svinoy and Jan–Mayen transects); B (between Jan-Mayen and Bear Island); C (between the transects Bear Island – Sorkapp); D (the transects Sorkapp and Fram strait). However, the trends and the interannual variability patterns are preserved (Fig. 3). More examples for different positions of the transects and variation of the reference temperature are presented in the Supplementary materials (Figure S1). Over the areas of the subregions A-D, the integral ocean-atmosphere and radiative heat-fluxes were also estimated. "

Figure S1. Examples of integral heat flux in AW layer depending on transects position and choice of reference temperature

[7]. Section 2.8: Vertical mixing or diapycnal mixing? Are you estimating the mixing at the base of the AW, which is along an isopycnal? If so, how robust is it to use vertical mixing coefficient Kz to estimate mixing across an isopycnal?

Response:

We estimate vertical mixing across the bottom isopycnal, so "diapycnal mixing" is the

right term. Isopicnals locally are almost parallel to the horizontal, typically inclined to the horizontal only by a few degrees, so vertical or diapycnal mixing practically give the same values. Therefore, it is commonly accepted to use the vertical mixing coefficient for estimating a cross-isopycnal mixing in the ocean.

[8]. Line 238-239: The correlation seems to result from the trend. What is the correlation after detrending the time series?

Response:

After detrending the correlation decreases slightly: for water temperature from 0.70 to 0.63, for U component from 0.48 to 0.44, for V component it doesn't change (remaining 0.60).

Added a sentence: Line 250: "The monthly mean current velocities, derived from the mooring data, also show a significantly higher variability compared to altimetry based ARMOR3D data. For annual mean U and V components, which are of the main interest for this study, correlations between the data-sets increase (in the presented example, from 0.5 to 0.7 and from 0.6 to 0.7, respectively). Removing long-term trends from the time series results in slight decrease in the correlations for the cross-flow U component (to 0.4), while does not change that for the along-flow V component and for water temperature."

[9]. Figure 5: Suggesting adding error bars to the time series plot.

Response:

Error bars were added for seasonal data, however, for annual mean time series plot this doesn't seem necessary as they simply reflect the strong seasonal (and subseasonal) variability, while making the plots difficult to read.

Figure 5. Validation of ARMOR-3D (blue) against in situ data at mooring F5 (red) located in the WSC at 78,5° N 6° E: a – water temperature (oC), b – zonal current velocity U (cm s-1) and c – meridional current velocity V (cm s-1). Left -Taylor diagrams

(ARMOR-3D is point B, in situ – point A), center – data time series, right - seasonal cycles. Data are averaged in 50-150 m layer.

[10]. Figure 6: Suggest adding uncertainties to the budget analysis. There is a clear difference of the mean meridional velocity between the ARMOR and the mooring (Figure 5), implying large uncertainties in the estimated advective heat flux. Again, the calculated heat flux is really a temperature flux, whose mean may be significantly modified with a different reference temperature.

Response:

Uncertainties of the means (in TW) were added to Figure 6.

Figure 6. Fluxes $\pm$ errors of the means (at the 95% confidence level).

[11]. Line 301: The correlation between Svinoy and Jan Mayen is as high as 0.7 according to Figure 7. Why is that a loss of correlation?

Response:

The correlation loss is due to atmosphere and eddies are removing heat from the Norwegian current and dispersing in across the Lofoten basin. The time variability of these processes do not necessarily correlate with that of the AW inflow across the Svinoy section. We are looking at the changes of the correlation coefficients between the transects as the AW progresses north, so "strong loss" of correlation refers to how fast correlation coefficient changes from transect to transect.

The description Lines 330-335 were changed to:

"The strongest correlation loss is found between Voring and Jan Mayen sections, while another one is between Isfjord and Fram sections. The correlation loss between Voring and Jan Mayen sections along the NwAC can be explained by an exceptionally high ocean eddy dynamics, which are effectively generated west of the Lofoten Islands (Isachsen, 2015) and redistributes the incoming heat over the area of the Lofoten basin,

further released to the atmosphere (Dugstad et al., 2019; Raj and Halo, 2016). We may expect somewhat similar reasons for the correlation loss between Isfjord and Fram sections (von Appen et al., 2015, Bashmachnikov et al., 2020). "

[12]. Figure 9c: There seems to be an increase of the dominant period with time. For example, in months 192-288, there is a dominant period of >84 months. Is there an explanation for that?

Response:

The time change of the period is presumably related to the corresponding changes in the oceanic heat advection entering the study region (possibly linked to the variability of AMO, NAO or EA patterns). However, we concentrated here on the variability of the heat fluxes within the region (from south to north), and the mechanisms of variability of the fluxes entering the region is already out of the scope of the present analysis.
* * *
[Figure]

[Figure]

**Fig. 1.** Figure 1.Schematic map of oceanic circulation in the study region

**Fig. 2.** Figure S2. Ocean atmosphere heat fluxes for region A, calculated using different datasets

[Figure]

**Fig. 3.** Figure S1. Examples of integral heat flux in AW layer depending on transects position and choice of reference temperature

**Fig. 4.** Figure 5. Validation of ARMOR-3D (blue) against in situ data at mooring F5 (red) located in the WSC at 78,5° N 6° E

[Figure]

**Fig. 5.** Figure 6. Fluxes ± errors of the means (at the 95% confidence level).

---

## Author Comment (AC2) · 5 Mar 2021

The authors investigated the heat flux in several sections northward in the Nordic Seas along the pathways of Atlantic Water into the Arctic. This was done using the AR-MOR3D dataset that derives from in-situ temperature and salinity data and satellite data. The heat flux variability is discussed in relation to atmospheric forcing that includes different weather regimes and North Atlantic Oscillation. The topic is of high scientific interest and this work could contribute to our understanding of the heat transport toward the Arctic. However, it is not clear what is the new findings in this paper. This should be more highlighted and discussed in the paper. The paper is well structured but the discussion should be improved (as mentioned above) and there are also too many errors or unexplained (or not shown) statements in the paper (see my comments below) to be accepted for publication.

**Response:**

Thank you for your comments and well-deserved critique. We added more thorough discussion to the paper, all shortcomings were addressed and missing information added. More detailed answers are added below.

**Detailed comments**

[1] Since the ARMOR3D and mooring observation have monthly values and there is strong seasonality in both data sets, there is not unexpected that the correlation between them is relatively high (Line 231-245 and figure 5). To compare the two datasets for interannually variability the seasonal signal should be removed before the correlation analysis.

**Response:**

It is true that seasonality plays significant role, however, as in this study we are looking both at the seasonal and interannual variability it was important to see how good reanalysis aligns with in-situ data on both time-scales. For study of the interannual variability the seasonal signal was naturally removed. We used two methods. As the first method, we averaged data to annual means (in figure 5 one can find both monthly and yearly values). For U and V components correlation coefficients between ARMOR and the mooring data increase slightly compared to the monthly means, as the monthto-month variability, strong in the mooring data, was smoothed.

**Added a sentence:**

Line 250: "The monthly mean current velocities, derived from the mooring data, also show a significantly higher variability compared to altimetry based ARMOR3D data. For annual mean U and V components, which are of the main interest for this study,
correlations between the data-sets increase (in the presented example, from 0.5 to 0.7 and from 0.6 to 0.7, respectively). Removing long-term trends from the time series results in slight decrease in the correlations for the cross-flow U component (to 0.4), while does not change that for the along-flow V component and for water temperature."

[2] Line 56-58: Several currents are mentioned but the East Iceland Current transports Arctic water and what is the West Icelandic Current? the (North-Icelandic) Irminger Current seems to be more relevant. Also, what is the Norwegian Atlantic Coastal Current? The location of the most central currents should be included in the map (Fig. 1.).

Response:

Authors corrected names of some of the currents. The phrase was changed to:

"The Faroe, East Icelandic and West Icelandic currents merge together (Fig.1) and carry a total of about 8–9 Sv into the Norwegian Current, including both, the NwAFC and the NwASC (Dickson et al., 2008; Rossby et al., 2017)."

Figure 1 was modified

[3] Line 90: What stands ARMOR3D for? There are many abbreviations without explanation (e.g. ARMOR3D, CMEMS, NACLIM, CLIMODE, COARE, ...).

Response:

Explanations for abbreviations were added to the text: A 3D multi-observations T,S,U,V product of the ocean (ARMOR3D) dataset, Copernicus Marine Environment Monitoring Service (CMEMS), North Atlantic Climate: Predictability of the climate in the North Atlantic/European sector (NACLIM) project, European Centre for Medium-range Weather Forecasts (ECMWF), Coupled Ocean–Atmosphere Response Experiment (COARE), CLIVAR (climate variability and predictability) and mode water Dynamics (CLIMOD), Marine Boundary Layer (MBL), Coupled Boundary Layer Air-Sea Transfer (CBLAST) experiments

OSD
[4] Line 119: "Vangengeim suggested...". Reference is missing.

Response:

Reference was added to the list: Vangengeim, G.Y. Application of Synoptic Methods to the Study and Characterization of Climate. Izvestia GGO, 2, 3-16, 1933 (in Russian)

[5] Line 130: "Vangengeim – Giers classification..." What is this classification?

Response:

A more comprehensive description of the referenced classification was added to the Supplementary materials, in the manuscript the link to Supplementary material. The phrase is added to the main text: "More in depth information about Vangengeim – Giers classification is provided in the Supplimentary materials."

""Vangengeim – Girs" classification is based on the analysis of hybrid-kinematic maps (Huth et al., 2008). The process of building a hybrid-kinematic map includes: registering the centers of cyclones and anticyclones, as well as positions of linear-like depressions and ridges from the daily synoptic pressure charts; drawing the demarcation line between the areas with high concentration of cyclones and depressions, and the areas with a high concentration of anticyclones and ridges. To reproduce kinematics of this process, trajectories of the baric formations are traced.

In 1933, Vangengeim suggested a set of indices characterizing atmospheric circulation. He introduced the concept of an elementary synoptic process (ESP). ESP is an evolution of the atmospheric pressure field during which the geographic distribution of the sign of the pressure anomalies and the direction of the main air transports are preserved within the Atlantic-European sector. All ESP could be further clustered in three main types of atmospheric circulation patterns: the western (W), the eastern (E) and the meridional (C) circulation types.

The description is based on Barashkova et al., 2015.

OSD
The particular feature of the western type (W) is the existence in the troposphere of waves with a relatively small amplitude moving fast from the west to the east. The baric features also move eastwards: cyclones – in the polar and mid-latitudes, anticyclones – in the subtropics. The high-pressure belt in the subtropics and the low pressure belt further north are well pronounced. This configuration of the atmospheric pressure results in predominantly zonal atmospheric transport. The meridional air-mass exchange weakens and negative temperature anomalies are observed in the polar regions (radiative cooling), positive – in tropical region (radiative warming).

The meridional type (C) is characterized by large amplitude waves in the troposphere. The northwards transport of the warm air along the western part of the ridges (to the Arctic), and the southwards transport along the eastern side, leads to high temperature contrasts, convergence of the high-altitude winds and dynamically linked growth of the sea-surface pressure. Areas of high temperature contrasts are favourable for formation of fronts and an enhanced cyclonic activity. During the circulation type C, the Icelandic lows is practically nonexistent due to a development of the high-pressure anomaly over the north Atlantic, the so called Atlantic Ridge. Further east, the Siberian Anticyclone strengthens and becomes connected with the Polar Anticyclone.

Similar to type C, the eastern type (E) is characterized by the tropospheric waves of large amplitude. However, the localization of ridges and troughs, as well as the distribution of the temperature anomalies, change to the opposite. Islandic low is now well developed. The Scandinavian Ridge is formed, while the winter Siberian anticyclone weakened and shifted west.

Huth, R., Beck, C., Philipp, A., Demuzere, M., Ustrnul, Z., Cahynová, M., KyselÃi, J. and Tveito, O.E., Classifications of atmospheric circulation patterns: recent advances and applications. Annals of the New York Academy of Sciences, 1146(1), pp.105-152., 2008

Barashkova N.K., Kuzhevskaya I.V., Polyakov D.V. Classification of forms of atmo-
spheric circulation: textbook. Tomsk: Publishing house of Tomsk University, 2015. (in Russian)"

[6] Line 151: The depth interval and the reference temperature for the heat fluxes are missing.

Response:

The caption for Figure 3 was changed to:

"Figure 3. a - interannual and b - seasonal variability of the heat fluxes in the AW layer (ref  $T = 0^{\circ}C$ ) across Jan Mayen section computed for different positions of the western and the eastern boundaries of the study region: blue – the section extends from the Scandinavian coast to the Jan Mayen island, red – the section starts at the Scandinavian shelf break goes west up to the western edge of the NwAFC jet (estimated as the first minimum of the modulus of the current velocity west of the jet), see Figure 2)"

[7] Line 160-162: "...even a small change in the position of the transect can lead to a significant change in the integral flux through the section.... These uncertainties must be taken into account when calculating balances within the studied areas" I cannot see that this has been done or have been discussed.

Response:

As there is almost an infinite amount of variations of the transects positioning, it is impossible to give a precise estimation of the uncertainties connected to this issue. We have performed a number of experiments, some of which are presented in the Figure below (added to the Supplement). The results show that, though sometimes affecting the absolute values of the fluxes, the variations of the transect positions or the parameters practically do not change neither the character of the interannual variability, nor the long-term trends, which are the focus of this study. This means that the results of the correlation drops and the character of the cycles discussed in this study are robust, independent of the variations in the section shapes described above.

OSD
Lines 168-176 were changed to "The heat fluxes through the western boundaries of the regions are most challenging to calculate with sufficient precision. The instability of the NwAFC, a relatively large (monthly) period of data averaging, the medium resolution of the available data, and anaccounted ageostrophic component can lead to a significant change in the integral flux through the section even with a relatively small change in the position of the transects. These uncertainties are taken into account when discussing the values of the ocean heat convergence in the subregions, limited by the transects: A (limited by the transects Svinoy and Jan–Mayen transects); B (between Jan-Mayen and Bear Island); C (between the transects Bear Island – Sorkapp); D (the transects Sorkapp and Fram strait). However, the trends and the interannual variability patterns are preserved (Fig. 3). More examples for different positions of the transects and variation of the reference temperature are presented in the Supplementary materials (Figure S1). "

Figure S1. Examples of integral heat flux in AW layer depending on transects position and choice of reference temperature

[8] Line 188-189: The authors use potential density thresholds for the definition of AW. Table 2 seems to be unnecessary and has several errors: Mork and Blindheim (1999) does not exist, Orvik et al. 2001 only studied the Svinøy section and not the whole Norwegian Sea (the ref. is also missing), Orvik and Niiler (2001) used 30 cm/s currents as AW pathways (not as definition of AW), reference of Furevik et al (2007) is missing, etc.

Response:

Following the reviewer comment we have corrected a few errors in the table: Mork and Blindheim (1999) is changed to Mork and Blindheim (2000), for Mork and Blindheim (2000) and Orvik et al. (2001) information that criteria was given based on Svinoy section was added, Orvik and Niiler (2001) was removed, missing reference of Furevik et al (2007) was added to the reference list. However, we disagree that this table is
unnecessary. The table gives a first comprehensive information on a variety of the criteria used for studying the same phenomena by different authors. The information is also used for our choice of the lower limit of the AW layer, and justifies the choice of different isopycnals along the Aw pathway north.

[9] Line 205-208: The reason for why dz=100 m was chosen is lacking? "dT is the temperature differences between the lower boundary of AW and surrounding waters" Is the surrounding waters 100 m (=dz) below the boundary of AW?

Response:

dz = 100 m was chosen simply because at the depths of few hundreds of meters the vertical layers in ARMOR3D dataset have spacing of 100 meters.

The phrase is changed to: "dT is the temperature differences between the lower boundary of AW and the first level below the selected AW limit"

[10] Line 209-229: I don't see that this calculation is necessary. Why not just use the constant value from Fer et al. with the reference. The uncertainties of Kz is probably less important than compared to other variables (e.g., dT and dz) or the chose of the reference temperature in eq. 1.

Response:

Kz varies in a wide range depending on the regional water dynamics and the stratification. Fer et al. presents estimations of Kz only for a specific region of the Lofoten basin. This value might not be representative for broader areas. To verify this we also estimated Kz via its well-known dependence on the Richardson number used in a variety of numerical models.

[11] Line 286-288: "The imbalance account to 10-20% ...that reflect the warming of AW...We should take into account uncertainties..." I cannot see that the uncertainties are discussed in the paper (e.g., what will be the uncertainties), and if this imbalance reflects the warming, how much will this be (in temperature/heat). This should then be

OSD
compared with other works.

Response:

Figure 6. was corrected and uncertainties were added to the plot

Figure 6. Fluxes  $\pm$  errors of the means (at the 95% confidence level).

Estimations of imbalances are now presented in the panels, however, as it was stated previously, the values, may vary depending on multiple factors (position of transect, reference temperature, depth of AW layer and etc.). This also forms problems in comparison of the integral fluxes presented with other studies. In particular one cannot compare such estimated with those of the regions with different vertical sections of depth limits. To our knowledge, the similar complete estimate of the balance of the heat fluxes in the suggested boundaries have not been performed. The heat imbalances are presented in the centers of the plots.

[12] Line 300: "...dropping to insignificant levels...". What is the significant level?

Response:

Significance of the correlation coefficients was tested with p-value (p > 0.05), which is the 95% significance level. The significant coefficients are now highlighted in the table 2.

[13] Line 301-303: That the correlation drops between Svinøy and Jan Mayen sections might also be due to the AW lies deeper and has longer residence time in the Lofoten Basin (e.g., Nilsen and Falck, 2006; Skagseth and Mork, 2012).

**Response:**

Thank you for pointing this out. How we see the story, the principle oceanic heat flux across the Jan Mayen section rather results from the direct water advection from the Svinoy by the two branched of the NwAC along the topographic guides (continental slope and the underwater ridge chain), rather than water entering the central basin and
**Printer-friendly version**

then being re-trapped by the current to continue its journey north. The long residence time would rather result in a stronger heat loss from the central parts of the basins due to ocean-atmospheric exchange and the vertical mixing.

Lines 330-335 were changed to:

"The strongest correlation loss is found between Voring and Jan Mayen sections, while another one is between Isfjord and Fram sections. The correlation loss between Voring and Jan Mayen sections along the NwAC can be explained by an exceptionally high ocean eddy dynamics, which are effectively generated west of the Lofoten Islands (Isachsen, 2015) and redistributes the incoming heat over the area of the Lofoten basin, further released to the atmosphere (Dugstad et al., 2019; Raj and Halo, 2016). We may expect somewhat similar reasons for the correlation loss between Isfjord and Fram sections (von Appen et al., 2015, Bashmachnikov et al., 2020)."

[14] Line 306-307: "...the cross-correlation analysis suggests the maximum correlations at zero time lag". Is this done?

Response:

Cross-correlation analysis has been performed, but as it hasn't provide us with any new information as the maximum positive cross-correlations were always obtained at zero time lag (see Figure S3 below).

Figure S3. Example of the cross-correlation's results (lag in months) for pairs Svinoy – Jan Mayen sections and Svinoy – Fram sections

[15] Line 381-382: "... is mostly shaped by the variations in the current velocity..." This should then be shown.

Response:

This sentence was removed

[16] Line 385-387: Could the 10-20% of the incoming heat be due to reduced air-sea
heat loss? See also my comments on Line 286-288.

Response:

Variations of air-sea heat fluxes are already taken into account, while the eddy transport is not directly addressed in this study. However, other studies (ex.Raj et al., 2020; Bashmachnikov et al, 2020) suggest that eddies may be responsible for this.

[17] Figure 1: the position of NwAFC is wrong.

Response:

Figure 1 was modified to add more information and correct errors

[18] Figure 2: Are the currents from ARMOR3D? At which depth? Is it same as for temperature (50m)?

Response:

Yes, mean currents are computed for the same depth-level (i.e. 50 m). Caption for the figure was changed to: "Figure 2. ...Black arrows indicate the mean currents at 50 m...."

[19] Figure 7. The correlation should also be done with time lags. While the velocity might have no or short time lag the temperature might have 1-2 year time lag (see e.g., Holliday et al., 2008; Skagseth et al., 2008; Chafik et al., 2015) which was also mentioned by the authors.

Response:

Addressed in the response to comment [14]

[20] Figure 9. The figure with wavelet amplitudes seems to be wrong. The amplitude should only be positive. Why is the heat flux integrated to 500 m? Why not use the boundary level of AW defined in the paper?

Response:
The word "amplitudes" was incorrectly chosen when in fact the color-scale represent normalized values of the wavelet coefficients, the caption for the figure was corrected. 500 meters in majority of cases represents AW layer, but it's true that this part of the caption was misleading, since all computations were done for AW layer. This was corrected too.

Figure 9. Time series (a, b) and wavelet diagrams (c, d) of interannual variations of heat fluxes: on the left – Svinoy section, on the right – Fram section, e - cross-wavelet diagram between the Svinoy and Fram sections.

[21]Table 2. What is the significant level

Response:

Significance of the correlation coefficients was tested with p-value. Caption changed to: "Table 2. Correlation coefficients of heat fluxes with climatic indices (bold italic – significant values, p > 0.05)"

OSD

---

## Author Comment (AC3) · 5 Mar 2021

Vesman and co-authors seek to investigate the variability in transport of heat through the Nordic seas towards the Arctic Ocean using a gridded dataset of monthly temperature, salinity and velocity fields derived from in situ and satellite data, and to frame this variability in terms of atmospheric forcing of the ocean. An improved understanding of heat transport in this region would be valuable, given its significance for changing ice cover in the Arctic in the coming years. The paper is clearly structured, and well written. It is, however, difficult to assess the significance of the results presented here in the absence of any discussion of the errors associated with them, which the authors

acknowledge are likely to be significant. The authors have attempted to provide some explanation of the physical basis behind the correlations in variability in heat fluxes that they see at various locations along the path of Atlantic Water (AW), but unfortunately this is unconvincing because the patterns of atmospheric forcing in terms of weather types that they present are inconsistent with their results. I offer some more detailed comments below, but I suggest that these shortcomings need to be addressed before the manuscript can be considered for publication.

Response:

Thank you for your comments and suggestions, we've rewritten discussion part to provide more information on atmospheric forcing and focused more on explaining the mechanisms behind connection with some weather types. Discussion of errors was also included.

Specific comments:

1. Line 123: I am not familiar with this particular categorisation. Can you provide some background – how it is derived, and why it is an appropriate description of the atmospheric patterns seen over the Nordic seas – for the benefit of readers who are unable to follow the Russian language references?

Response:

A more comprehensive description of the referenced classification was added to the Supplementary materials, in the manuscript the link to Supplementary material. The phrase is added to the main text:

"More in depth information about Vangengeim – Giers classification is provided in the Supplimentary materials."

""Vangengeim – Girs" classification is based on the analysis of hybrid-kinematic maps (Huth et al., 2008). The process of building a hybrid-kinematic map includes: registering the centers of cyclones and anticyclones, as well as positions of linear-like depressions and ridges from the daily synoptic pressure charts; drawing the demarcation line between the areas with high concentration of cyclones and depressions, and the areas with a high concentration of anticyclones and ridges. To reproduce kinematics of this process, trajectories of the baric formations are traced.

In 1933, Vangengeim suggested a set of indices characterizing atmospheric circulation. He introduced the concept of an elementary synoptic process (ESP). ESP is an evolution of the atmospheric pressure field during which the geographic distribution of the sign of the pressure anomalies and the direction of the main air transports are preserved within the Atlantic-European sector. All ESP could be further clustered in three main types of atmospheric circulation patterns: the western (W), the eastern (E) and the meridional (C) circulation types.

The description is based on Barashkova et al., 2015.

The particular feature of the western type (W) is the existence in the troposphere of waves with a relatively small amplitude moving fast from the west to the east. The baric features also move eastwards: cyclones – in the polar and mid-latitudes, anticyclones – in the subtropics. The high-pressure belt in the subtropics and the low pressure belt further north are well pronounced. This configuration of the atmospheric pressure results in predominantly zonal atmospheric transport. The meridional air-mass exchange weakens and negative temperature anomalies are observed in the polar regions (radiative cooling), positive – in tropical region (radiative warming).

The meridional type (C) is characterized by large amplitude waves in the troposphere. The northwards transport of the warm air along the western part of the ridges (to the Arctic), and the southwards transport along the eastern side, leads to high temperature contrasts, convergence of the high-altitude winds and dynamically linked growth of the sea-surface pressure. Areas of high temperature contrasts are favourable for formation of fronts and an enhanced cyclonic activity. During the circulation type C, the Icelandic lows is practically nonexistent due to a development of the high-pressure anomaly over

the north Atlantic, the so called Atlantic Ridge. Further east, the Siberian Anticyclone strengthens and becomes connected with the Polar Anticyclone.

Similar to type C, the eastern type (E) is characterized by the tropospheric waves of large amplitude. However, the localization of ridges and troughs, as well as the distribution of the temperature anomalies, change to the opposite. Islandic low is now well developed. The Scandinavian Ridge is formed, while the winter Siberian anticyclone weakened and shifted west.

Huth, R., Beck, C., Philipp, A., Demuzere, M., Ustrnul, Z., Cahynová, M., Kyselá, J. and Tveito, O.E., Classifications of atmospheric circulation patterns: recent advances and applications. Annals of the New York Academy of Sciences, 1146(1), pp.105-152., 2008

Barashkova N.K., Kuzhevskaya I.V., Polyakov D.V. Classification of forms of atmospheric circulation: textbook. Tomsk: Publishing house of Tomsk University, 2015. (in Russian)"

2. Figure 2: The maps are small and the detail difficult to make out, but it looks as if you might be losing some of the northward AW flow and periodic southward recirculation at the eastern end of some of your transects, particularly at the southern end of the Barents Sea Opening. The current is strong here, and its position varies a fair amount. (See, for example, Wang et al. 2019.) You mention at Line 160 that your results are sensitive to the position of the transects in relation to the western boundaries of your regions, but do not, so far as I can see, provide any quantification of the uncertainties associated with choice of position.

Response:

As there is almost an infinite amount of variations of the transects positioning, it is impossible to give a precise estimation of the uncertainties connected to this issue. We have performed a number of experiments, some of which are presented in the

[Figure]

Figure below (added to the Supplement). The results show that, though sometimes affecting the absolute values of the fluxes, the variations of the transect positions or the parameters practically do not change neither the character of the interannual variability, nor the long-term trends, which are the focus of this study. This means that the results of the correlation drops and the character of the cycles discussed in this study are robust, independent of the variations in the section shapes described above.

Lines 168-176 were changed to "The heat fluxes through the western boundaries of the regions are most challenging to calculate with sufficient precision. The instability of the NwAFC, a relatively large (monthly) period of data averaging, the medium resolution of the available data, and anaccounted ageostrophic component can lead to a significant change in the integral flux through the section even with a relatively small change in the position of the transects. These uncertainties are taken into account when discussing the values of the ocean heat convergence in the subregions, limited by the transects: A (limited by the transects Svinoy and Jan–Mayen transects); B (between Jan-Mayen and Bear Island); C (between the transects Bear Island – Sorkapp); D (the transects Sorkapp and Fram strait). However, the trends and the interannual variability patterns are preserved (Fig. 3). More examples for different positions of the transects and variation of the reference temperature are presented in the Supplementary materials (Figure S1). "

On the transects in the shelf: we have also done the computation including the transects on the shelf. However, as the ARMOR currents are based on the sea-surface altimetry, the altimetry is not reliable at a distance less than 50 km to the coast. So we did not use the points which are closer than 50 km to the coast.

3. Line 204: "the base of the upper layer". Is this the AW layer?

Response:

Yes, the base of the upper layer is the base of the AW layer, to make it more clear the line was changed to: "…flux through the base of the AW layer is…"

4. Line 232: "we compare the statistical properties of all available mooring observations. . ..with those in the nearest grid-point of the ARMOR3D dataset." Would you see a better correlation if you compared the mooring observations with interpolated values from the ARMOR3D dataset? LaCasce 2005 found low spatial correlations between current meter readings taken from moorings only a few kilometres apart on the Norwegian Slope, and similar lack of correlation might be expected between current meter observations and ARMOR3D grid points over a similar distance. Nevertheless, the spatial resolution of satellite observations underlying the gridded dataset might be insufficient for further interpolation to offer an improvement.

Response:

Majority of moorings are situated very close to the ARMOR grid points. Comparison was done also using ARMOR results interpolated to the mooring positions. The results were practically the same (see figure below). Overall ARMOR currents, based on extrapolation of the altimetry currents down using the thermal wind relation naturally smooth the space-time variability the current velocity.

Fig.2 Comparison of ARMOR data obtained using interpolation and using closest grid point

5. Line 236: "data are binned to 100 m vertical bins". Why 100 m?

Response:

Moored instruments change their vertical position in time (see Figure below) due to ocean dynamic (currents, storms etc.), as well as during re-deployment of the moorings (which results in a slight change of the mooring positions). Analyzing the positions of the instruments during research period, we concluded that 100 m vertical bin of ARMOR3D data covers practically all possible changes in the instrument positions around the upper, middle and lower water levels, and can be used for comparison with the variability of the moored time series over the whole period of observations.

Fig.3. Changes in the depth of moored instruments with time

6. Line 240: "current velocity...derived from ARMOR3D shows lower…...variability, compared to in situ data". Variability will depend on the scale over which values are averaged. The mooring data are collected at fixed points, so one might expect them to exhibit higher variability.

Response:

That is true (see also our answer to comment 4 above), but still worth mentioning as the whole paragraph is dedicated to comparison of the datasets.

7. Line 259: The NwACC carries fresher water of Baltic origin, not AW. (Skagseth et al. 2008, in your reference list.

Response:

Thank you for correction. The corresponding information is missing in Skagseth et al. 2008, but we found it in Gascard and Mork (2008) in the same book. This is added to the reference list sentence was changed to:

"The heat advection across the section is split between three main cores of the warm waters: the coastal branch at 10° E (NwACC) that carries a fresher water of the Baltic origin, further affected by the freshwater runoff off the Norwegian coast (Gascard and Mork, 2008), the slope branch between 5 and 6° E (NwASC) and the polar frontal branch between 2 and 3° E (NwAFC)."

8. Lines 261/2: Are these long term mean heat fluxes?

Response:

Yes, these are long term means. Specification added:

Line 279: "Our analysis shows that the largest mean (over the study period) heat flux is directed northward along with the NwASC."

9. Figure 6: Balances can only be given to the precision of the least precise of the inputs, not to 0.1 of a terawatt. But more seriously, what are the error estimates for these calculations?

Response:

Figure was corrected and uncertainties are added to the plot

Figure 6. Fluxes ± errors of the means (at the 95% confidence level).

10. Figure 8. Are these mean sea level pressure fields, as the caption says, or anomalies? (The values on the colour bar are too small to be legible.) If you wish to relate variability in ocean transports to variability in atmospheric forcing, do you not wish to look at the anomalies from long term mean?

Response:

We agree with the reviewer that the anomalies may often be more clear. We replaced the pressure patterns with anomalies of the wind stress curl, averaged over the corresponding wind patterns of each of the weather types. However, we think that anomalies of the wind vector will not provide the necessary information, as they often do not reflect the direction of the real wind, important for the discussion.

Figure 8. Anomalies of the wind stress curl (red – increase of the sea level, blue – decrease of the sea level in meters), dominant wind patterns (vectors) over the North Atlantic associated with circulation types: a - W, b - C and c – E, dashed lines – bathimentry, red arrows – AW pass

11. Lines 331-3: This doesn't seem quite right. The colour shading in Figure 8a. obscures the wind vectors, but they appear to point slightly northwards along the coast at Svinøy, but offshore at Spitsbergen. So we might expect to see some build up of sea level against the coast and consequent enhancement of the shelf current at Svinøy, but no similar Ekman effect at the more northerly transects.

Response:

The reviewer discusses here only one effect of the wind – a sea-level change due to the Ekman convergence/divergence near the cost. However, there is also another effect: Ekman convergence below the anticyclonic wind stress curl and divergence below the cyclonic one. This is particularly important for the NwAFC, but also should be considered for the NwASC, as the shelf break is often relatively far from the coast. We added the following text discussing these results in detail:

Lines 363-391:

"Along with the sea-level drop/increase near the coast, which depends on the direction and intensity of the along-coast wind component, we consider convergence/divergence of the Ekman flux in the open ocean (Ekman pumping), which is proportional to the wind stress curl. In the first case, the sea-surface vertical velocity can be estimated as w= $\tau$/fL, while in the second case Ekman related at the sea-surface w= - 1/f rot($\tau$), where $\tau$ is the wind stress curl, $\tau$ is the wind stress curl component along the coast, is the mean water density, f is the Coriolis parameter and L is the distance from the coast. We are interested in the anomalies of the vertical velocity relative to the climatic mean wind fields associated with weather types W, C, E (Fig. 8). Acceleation/deceleration of the currents are formed by changes in the sea-level gradients across the axis of the branches of the Norwegian Current, forced by the wind fields characteristic for a particular weather type. In Figure 8, changes in the sea-level for each of the weather types relative the climatic mean state are presented as the vertical velocity anomalies, the gradients of which are of the main interest below.

Along the Norwegian coast, the acceleration of the along-shore branch of the Norwegian Current due to the sea-level build-up (forced by the southwesterly winds) is expected for weather types E (Fig.8c) and W (Fig.8a), but not for type C (Fig.8b). For type W, the anticyclonic wind-stress curl also results in an anomaly of the Ekman pumping convergence along the Norwegian shelf and over the Voring plateau. The same is

observed along the continental slope west of the Barents Sea Opening. This further increases the sea-level build-up east of the NwASC all the way to 75°N, maintaining a higher current velocity and a stronger heat advection. The opposite tendency is observed for types E and C, diminishing the effect of the near coast sea-level build-up for type E, or enhancing the negative near-coast sea-level anomalies for type C. Further north of the Norwegian shelf, type C favours a stronger warm AW outflow into the Barents Sea, while the opposite situation is observed for type W (and E).

West of Spitsbergen, for type W, the clear positive effect of the wind-stress curl on the NwAFC and the WSC transport is observed at 79°N, while further south an acceleration of the NwASC may be compensated by a deceleration of the WSC. For weather type E, an acceleration along the southern part of the island is accompanied by a deceleration further north. For weather type C a clear deceleration of both, the WSC and the NwASC is governed by a northeastwards sea-level drop (i.e. a northeastwards increase of the negative vertical velocity forced by Ekman pumping).

In summary, the analysis above suggests that the Ekman pumping forced by the wind stress curl, together with a near-coast sea-level build-up (mostly along the Norwegian coast), should increase the northward current velocity practically along all its path through the Nordic seas for weather type W and decrease – for weather type C. For weather type E the current accelerations and the decelerations alternate along the current axis. With gentle winds and a relatively small variation of the Ekman pumping anomalies over the Nordic Seas, we do not expect a pronounced consistent increase or decrease of the current velocity along the northward pathways of the AW."

12. Lines 334-7: It is a convergence or divergence of Ekman transport in association with the coast that generates the sea level gradients which lead to the variability in the geostrophic slope current. Ekman transport in a southerly (along slope) direction cannot, therefore, decrease the NwASC, because it does not involve convergence or divergence of transport. It should have no effect.

Response:

Please see our responses to comments 10 and 11. Ekman pumping at the coast does not tell the full story of the Ekman convergence/divergence patterns.

13. Line 337: The along slope winds shown for Type E in Figure 8b. do not appear to be significantly weaker than those for Type W in the same region.

Response:

Please see our responses to comments 10 and 11. Ekman pumping at the coast does not tell the full story of the Ekman convergence/divergence patterns.

14. Line 352: "since 2005 it [the heat advection] started to decrease". Figure 9b. appears to show a recovery of heat flux in the final two years. Do we not just we decadal-scale variability here, rather than any trend?

Response:

Yes, we probably observe some decadal-scale variability. Any trend in a time-limited data might be a part of a variation with a period longer then the study period. During the time interval of our study, the recovery of the last couple of years doesn't change the long-term positive trend. The sentence was changed to: Lines 404-408:

"However, the heat advection across the Fram section increases only in the beginning of the 2000s. Since 2005 it startes decreasing, with some recovery in 2016-2017. Overall, no significant long-term trend is noted during the study period. Thus, despite the general increase in the water temperature in the south of the region, the northern sections do not demonstrate a positive trend in the heat fluxes during the latest decades. This is one of the factors reducing the correlations."

15. Line 358: How sensitive are these results to choice of transect? Would they show similar periodicity and coherence if you chose the Vøring and Isfjord transects, for example?

Response:

Overall results from all transects show the same dominating period, while the amplitudes of the cycles show a gradual reduction in amplitudes. In this sense, results from Svinoy and Fram sections are sufficiently representative for the transects in-between. Wavelet diagrams from all sections along latitudes are presented in the FigureS4, added to the Supplementary material

Figure S4. Wavelet diagrams of interannual variations of heat fluxes

Technical corrections

All figures: small text is difficult to read. Can you make the labelling clearer? Colours are also difficult to distinguish in Figure 7.

Response:

We tried to make figures more clear, colors were slightly adjusted

Line 141: "the currents are strongly bottom trapped". Do you mean "topographically steered"?

Response:

Corrected (Line 296)

Line 268: "stronger northerly winds". This English expression is commonly understood to refer to winds blowing from the north, whereas I think you are talking about winds blowing from south to north. I suggest "northward-blowing winds".

Response:

Changed to "northward-blowing winds "

Line 272: Delete the comma after "heat flux". It changes the meaning of the phrase.

Response:

Changed to "On average over the study period (1993–2017), the major heat flux of 406 TW enters the Norwegian Sea across the Svinoy section."

Line 320: "into the Barents Sea"?

Response:

Changed to "into the Barents Sea" Line 351

Line 326: "the correlations go to zero". Go to zero, or just become small?

Response:

Changed to: "go to insignificant values close to 0" Line 346

[Figure]

**Fig. 1.** Figure S1. Examples of integral heat flux in AW layer depending on transects position
and choice of reference temperature

**Fig. 2.** Comparison of ARMOR data obtained using interpolation and using closest grid point

[Figure]

[Figure]

**Fig. 3.** Changes in the depth of moored instruments with time

[Figure]

**Fig. 4.** Figure 6. Fluxes $\pm$ errors of the means (at the 95% confidence level).

**(a)** Anomaly of Rot τ during weather type W

**(b)** Anomaly of Rot τ during weather type C

**(c)** Anomaly of Rot τ during weather type E

**Fig. 5.** Figure 8. Anomalies of the wind stress curl

[Figure]

**Fig. 6.** Figure S4. Wavelet diagrams of interannual variations of heat fluxes